# CONFORMAL REASONING: UNCERTAINTY ESTIMATION IN INTERACTIVE ENVIRONMENTS

## ABSTRACT

Effectively balancing uncertainty and decision-making is critical in real-world reasoning tasks. Split conformal prediction (SCP)—a popular statistical framework for uncertainty quantification that constructs prediction sets using a calibration set of interactions—is significantly limited in multi-turn interactive settings. In particular, collecting calibration data requires the interaction trajectories to be determined *a priori*, inducing a heuristic bias that weakens coverage. Moreover, SCP cannot leverage the model's own uncertainty to better guide its decision-making without breaking coverage guarantees. To address these limitations, we propose CONFORMAL REASONING, a novel offline conformal framework built upon adaptive conformal inference methods that 1) is robust to distribution shift, 2) allows for flexible score function design, and, most importantly, 3) leverages the prediction set into the model's future decision making, all while preserving marginal coverage guarantee. On three real-world tasks including medical diagnosis, embodied question-answering, and twenty questions, we show that conformal reasoning empirically achieves theoretical coverage guarantees while showing improved accuracy and efficiency.

## 1 INTRODUCTION

Large Language Model (Brown et al., 2020, LLM) agents are increasingly deployed for complex decision-making tasks in the wild (Bommasani et al., 2021). In particular, agents must frequently interact with different aspects of the environment. A crucial component of interactive decision-making is an agent's uncertainty over its actions. Robust uncertainty estimation provides a heuristic for agents to abstain from making a decision and instead gather more information; this process can continue iteratively until the agent is confident enough to make a decision, or might require human assistance. This balance between uncertainty and decision-making is essential in high-stakes settings, such as medical diagnosis (Li et al., 2024).

Traditional methods such as post-hoc calibration and prompting have been used to improve LLM confidence estimation (Geng et al., 2024), but these techniques lack the formal guarantees necessary for reliable abstention decisions. **Conformal prediction** (Shafer & Vovk, 2008; Angelopoulos et al., 2023) fills this gap by providing statistically valid prediction sets with formal confidence guarantees based on a user-defined threshold. This framework ensures that the correct answer is included within a set of plausible options with high probability, offering a principled approach to confidence estimation and has been proven effective in LLMs for multiple-choice tasks (Kumar et al., 2023).

However, existing applications of conformal prediction in *interactive scenarios* leave significant room for improvement. Traditional **split conformal prediction (SCP)** requires a fixed calibration set of trajectories to be determined upfront, each coming with an associated noncomformity score (Ren et al., 2023; 2024; Xu & Xie, 2021). This collection of a calibration set requires several design decisions that are difficult to make *a priori*. For instance, the trajectories may be collected until a policy—here, an AI agent interacting with its environment—concludes, or to a fixed time cut-off. Neither of these calibration approaches is robust to the intricacies of the interactive multi-turn setting. A policy-dependent approach that relies on a threshold for termination might be used for tractable collection of a calibration set; yet, in turn, the *conformal* threshold $\hat{q}_\alpha$ generates trajectories

---

*Equal contribution in alphabetical order.

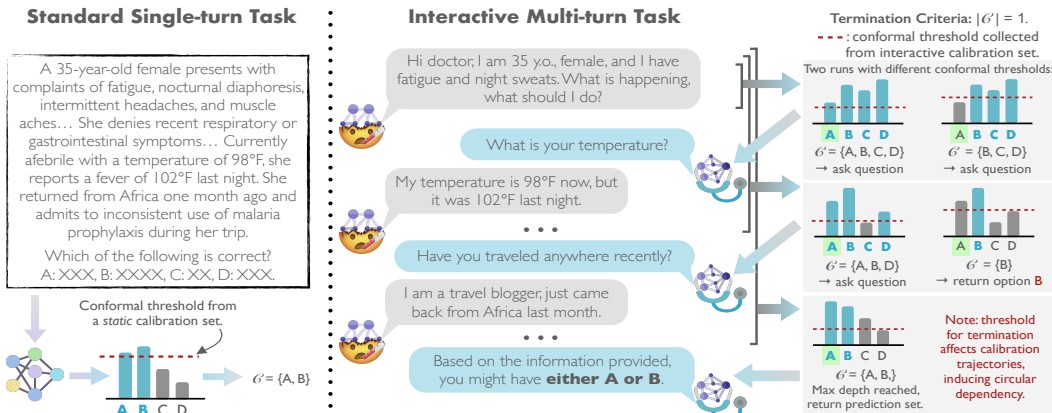

Figure 1: Single vs. multi-turn conformal prediction. In single-turn question-answering tasks (left), conformal prediction involves obtaining a conformal threshold from a fixed set of calibration points, providing robust coverage guarantees. However, in interactive multi-turn tasks (right), the conformal threshold collected multi-turn calibration trajectories is often subject to a maximum sequence length. This dependence on this initial condition induces heuristic bias, which breaks traditional split conformal prediction (SCP) coverage guarantees. Note that in the right trajectory, option A is eliminated in turn 1 but its probability exceeds the threshold in turn 2: our proposed Refresh Prediction technique can recover this option while standard set construction techniques cannot.

distributed differently from that of the calibration set, creating a circular dependency. On the other hand, a fixed time cutoff might create finite-sample in differences scores collected in a calibration set versus those seen at test time, inducing a *heuristic bias* that causes conformal guarantees to fail. Moreover, SCP prevents the conformal set—i.e., the measure of the agent's uncertainty—from being used in the decision-making policy; otherwise, the calibration set would also need to use conformal sets in its own decision-making policy, creating a circular dependency.

To address these limitations, we propose a new framework, CONFORMAL REASONING, which enables the use of conformal prediction in interactive reasoning tasks as demonstrated in Figure 1. CONFORMAL REASONING builds upon adaptive conformal inference (Gibbs & Candes, 2021, ACI), a variant of conformal prediction designed to construct valid prediction sets in response to arbitrary distribution shifts by updating its quantile parameter based on the outcome of each sample in a sequential manner. An ACI-based approach is able to preserve guarantees despite the circular dependency and heuristic bias issues in interactive setups. We propose a novel *train-test offline adaptation paradigm*, in which we can learn the quantile parameter from the train set and eliminate the need for sample-wise feedback at test time. Our novel application of ACI expands the design space of score and prediction set formulation and leverages the model's uncertainty to guide decision making. Overall, our approach introduces three key components that extend the capabilities of conformal reasoning for interactive tasks: 1) **Offline Adaptation (OA)** paradigm that applies ACI's robustness to distribution shifts to the multi-turn interactive setting without test-time sample-wise feedback, 2) expanding the OA framework with flexible prediction set construction, noted as **Refresh Prediction (RP)**, and 3) incorporating uncertainty into subsequent actions, noted as **Guided Reasoning (GR)**, all while preserving conformal guarantees.

We validate our method across three tasks: interactive medical diagnosis, embodied question answering, and open-ended twenty questions. Our results demonstrate that **Conformal Reasoning** significantly improves prediction accuracy, interaction efficiency, and the specificity of prediction sets compared to standard SCP, making it a practical and robust solution for real-world applications. To summarize, the key contributions of this work are:

1. We introduce CONFORMAL REASONING, a framework that adapts adaptive conformal inference (ACI) to interactive reasoning tasks, enabling models to dynamically adjust prediction sets during multi-turn interactions while maintaining robust coverage guarantees.

2. We propose two novel conformal techniques: **Refresh Prediction (RP)**, which refines prediction sets over the course of the interaction to improve the model's efficiency, and **Guided Reasoning (GR)**, which leverages the model's uncertainty in the next action to improve accuracy.

3. Our proposed *Init-Train-Test* paradigm—**Offline Adaptation (OA)**—is the first to eliminate the need for step-wise feedback during inference by learning a prediction threshold during training, simplifying deployment in real-world interactive systems where immediate correctness feedback may not be available.

4. We empirically validate our method across three challenging domains—**interactive medical diagnosis**, **embodied question answering**, and **open-ended twenty questions**—demonstrating significant improvements in prediction accuracy, interaction efficiency, and prediction set specificity compared to traditional split conformal prediction (SCP) methods.

## 2 RELATED WORK

**Interactive Reasoning and LLM Uncertainty Estimation.** The integration of LLMs into interactive systems has opened new avenues for complex decision-making tasks (Ouyang et al., 2022; Wang et al., 2023). These systems, however, often lack robust and calibrated uncertainty quantification, which are crucial for reliable decision-making. Desai & Durrett (2020) found that larger models are not necessarily better calibrated, highlighting the need for methods that provide reliable confidence estimates. Srivastava et al. (2022) and Geng et al. (2023) have shown that LLMs struggle at producing numerical confidence scores (Srivastava et al., 2022; Geng et al., 2023). To this end, other works have explored confidence calibration in LLMs via temperature scaling (Xie et al., 2024), prompting (Tian et al., 2023), and multi-agent deliberation (Yang et al., 2024). For high-stakes applications such as medical diagnosis, interactive reasoning becomes critical. Li et al. (2024) proposed MediQ, a framework that simulates clinicians interactively gathering patient information to make accurate diagnoses; there, better confidence estimation leads to higher diagnostic performance.

**Online Conformal Prediction.** Methods in adaptive conformal prediction were originally designed to provide coverage in adversarial online settings. The primary method for adapting to online distribution shifts is via online gradient descent, first proposed in Gibbs & Candes (2021); successive work has similarly used online subgradient updates with respect to tracking the quantile (Angelopoulos et al., 2024a; Bhatnagar et al., 2023; Angelopoulos et al., 2024a). Many of these works have focused on responsiveness to distribution shifts and characterizations of validity of prediction sets, such as bounds on measures of regret and techniques from online learning (Gibbs & Candès, 2024; Bastani et al., 2022; Jung et al., 2022; Zhang et al.). Our work builds upon existing approaches in online conformal prediction, but applies these techniques in a fundamentally different setting: our approach applies an online algorithm to data collected offline, allowing for adaptation to distributional heterogeneity in the calibration set while learning a quantile that attains marginal coverage.

**Conformal Prediction in Interactive Environments.** Conformal prediction has been used for trajectory prediction and planning within dynamic environments; these approaches generally leverage online technique (Gibbs & Candes, 2021; Gibbs & Candès, 2024; Angelopoulos et al., 2024b) or attain marginal coverage over the distribution of trajectories (Xu & Xie, 2021; Stankeviciute et al., 2021; Zaffran et al., 2022). Alternatively, a related line of work has characterized conformal prediction under feedback shifts, largely through the perspective of reweighting under induced covariate shifts (Fannjiang et al., 2022; Prinster et al., 2024). Recently, split conformal prediction has also been applied for robot decision-making under uncertainty, using conformal prediction sets as indicators for human-in-the-loop correction or as exploration stopping times (Ren et al., 2023; Liang et al., 2024). Our work is most similar to Ren et al. (2024), though with notable exceptions: our method of online adaptation to offline data enables using techniques that are incompatible with the approaches used in Ren et al. (2024) but enable empirical improvement.

**Conformal Prediction and Abstention.** Recent work has explored using conformal-style techniques for abstention, which requires controlling uncertainty over the selective classification error (Angelopoulos et al., 2021; Gui et al., 2024). These approaches are generally framed through risk control, where with high probability the selective classification error is controlled below a user-selected threshold using a calibration set. While these approaches successfully control the selective classification error and thereby give guarantees (w.h.p.) of the form $\mathbb{P}(y \in C(x) \mid |C(x)| = 1) \geq 1 - \alpha$, in a naive implementation they do not provide meaningful prediction sets. In contrast, we hope to attain marginal coverage guarantees typical in the conformal prediction literature.

Our approach *interpolates* between Angelopoulos et al. (2021) and Ren et al. (2024) to achieve a correct $1 - \alpha$ coverage guarantee. Ren et al. (2024) requires the interactions to terminate at the maximum allowed timestep for their specific scenario; while not achieving conditional coverage,

this approach would provide a marginal guarantee at the expense of the efficiency. In contrast, the aforementioned risk control approaches ensure that terminating at a prediction set of size one would provide a valid conditional coverage guarantee at the cost (in a naive implementation) of not offering any terminal prediction set. In particular, both Angelopoulos et al. (2021) and Gui et al. (2024) compute conformal $p$-values for each test point and use these $p$-values to determine whether the model's prediction should be used and elicited or abstained from. Here, we empirically demonstrate that we *retain marginal coverage* for the final prediction set at termination time. Moreover, our approach offers a more efficient alternative to the approach of Ren et al. (2024) that goes to the maximum sequence length, as we enable the agent to either terminate at a prediction set of size one or offer a terminal prediction set that can be used to aid a human practitioner.

## 3 PROBLEM FORMULATION

Consistent with the notation used in Ren et al. (2024), we consider a distribution over scenarios, $\xi \sim \mathcal{D}$, for an agent to interact with. Each scenario, $\xi$, is a tuple $\xi := (e, y, T, x^0)$, where $e$ represents the environment, $y \in \mathcal{Y}$ is the ground truth categorical answer to the posed question, $T \in \mathbb{Z}_+$ is the maximum number of times the agent is allowed to interact with the the environment to find $y$, and $x^0$ is the agent's initial state. Our goal is to design an agent that sequentially interacts with the environment, $e$, to predict the hidden answer, $y$, within $T$ steps of interactions. We model the agent as a policy $\hat{\pi}(\cdot)$, that, at each step $t \in [T]$, takes as input the current state $x^{t-1}$, and then takes an action, $a^t$, drawn from the policy: $a^t \sim \hat{\pi}(x^{t-1})$. The environment provides a feedback, $e(a^t | a^1, \ldots, a^{t-1})$, based on the current action given all previous actions, which in turn contributes to updating the state, $x^t$, that is then fed back into the agent. This updated state represents the refined knowledge the agent has about the state of the environment and the answer. For example, a state $x^t$ can be the complete transcript of all interactions so far, i.e. $x^t = (x^0, a^1, e(a^1), a^2, \ldots, e(a^{t-1}))$.

To make this concrete, we describe three tasks (Table 1) that we consider in our later experiments:

**Proactive Medical Information Seeking (MediQ).** In medical diagnosis, a clinician typically begins with an initial complaint and gathers further information (e.g., patient history, symptoms, lab results) before reaching a diagnosis. We use the MediQ benchmark (Li et al., 2024), an interactive version of the MedQA dataset (Jin et al., 2021) that presents some initial information $x^0$ that is the patient's demographic information and the chief complaint, and a medical question $\kappa$ and tests a model's ability to take actions $a$ by asking relevant questions to the patient until it has enough information to make a diagnosis $y$.

**Embodied Question Answering (EQA).** The Embodied Question Answering (EQA) task assesses a model's ability to actively explore a 3D environment, gather visual and spatial information, and answer questions about the scene. We use the HM-EQA dataset (Ren et al., 2024), which focuses on human-robot scenarios within photo-realistic 3D indoor environments constructed from the Habitat Matterport 3D dataset (HM3D) (Ramakrishnan et al., 2021). The agent controls a robot to navigate a habitat $e$ starting from an initial position $x^0$ that is the robot's initial pose and current RGB image observed on the onboard camera. The dataset consists of questions requiring the agent to locate objects, understand room layouts, or explore areas that may not be immediately visible; an example is "Where did I leave my red jacket?" with possible answers $\mathcal{Y} =$ "a) On the couch, b) On the kitchen table, c) In my office, or d) On the coat rack." The agent can take actions $a$ to move the robot to a different location and observe a new image $e(a^t)$ from the environment $e$.

**Open Ended 20 Questions (20Q)** The 20 Questions task involves two roles: the Answerer, who selects a target word, and the Guesser, who asks Yes/No questions to identify the target. This interactive game evaluates the model's ability to ask strategic, information-seeking questions and make accurate guesses based on partial information. In contrast to MediQ and EQA, where the answer space is constrained, the 20 Questions task involves an open-ended answer space. We use two datasets for this task: (1) **Original**—a hand-curated dataset of 340 target words from Hu et al. (2024), which is sourced from the BIG-bench 20 Questions task (BIG-bench collaboration, 2021), the 20 Questions official website[1], and an object concept database (Hebart et al., 2019); and (2) **Auto20Q**—a larger dataset of 2,000 target words automatically generated using GPT-4, covering a broad range of object categories to simulate more varied and challenging interactions. The training, validation, and test sets are partitioned accordingly to ensure balanced coverage across categories.

---

[1]https://blog.prepscholar.com/20-questions-game

| Task | | MediQ (Li et al., 2024) | EQA (Ren et al., 2024) | 20Q |
|---|---|---|---|---|
| Scenario | $\xi$ | Clinical interaction | Assistant robot | Word guessing |
| Environment | $e$ | Medical inquiry | Household simulation | Objects (nouns) |
| Answer | $y$ | Multiple choice option | Multiple choice option | Target word |
| Max Turns | $T$ | 15 | 30 | 20 |
| Initial Info | $x^0$ | Age, gender, complaint | Pose, RGB image | Game description |
| **Example Step** | | | | |
| Question | $\kappa$ | `What is the most likely diagnosis for this patient?` | `What color is the curtain in the bedroom?` | `Is X a living thing?` |
| Incremental Info | $x^t$ | `She has vomited 3 times and progressively became more confused.` Elicited patient info |  Picture of living room | `No` Answerer response |
| Intermediate Reasoning | $\pi$ | `The confusion, with symptoms like vomiting, points to hypoglycemia, consistent with insulin overdose...` Clinical rationale |  Semantic value repr. | `Out of the remaining options, X1:fits desc.; X2:fits desc.; X3:doesn't fit desc. ...` Option set partition |
| Episode-level Prediction Set | $C^\tau(x^\tau)$ | A) Dumping syndrome **0.36** B) Insulin overdose **0.33** C) Malnutrition 0.26 D) Propranolol overdose 0.05 | A) Grey **0.56** D) Blue **0.13** B) Black **0.22** C) While 0.09 |  |
| Agent Action | $a^t$ | Ask follow-up question. E.g. `"What is your medical history?"` | Navigate to bedroom. | Ask follow-up question. E.g. `"Is X man-made?"` |

Table 1: Examples of MediQ, EQA, and 20Q task scenarios and their elements.

**Metrics.** At any $t_{\text{ans}} \leq T$, the agent can choose to stop interacting and output a set of answers $C \subseteq \mathcal{Y}$ that we call a *prediction set*. Given a target $\alpha$, our aim is to achieve $(1 - \alpha)$ marginal coverage as closely as possible, while using minimum number of queries , and aiming for minimum size prediction set. Concretely, we aim to optimize the following metrics:

1. Coverage $:= \mathbb{P}(y \in C)$; this evaluates the reliability of the prediction set. The goal is to ensure that the true answer is included with a probability of at least $1 - \alpha$, i.e., Coverage $\geq 1 - \alpha$.

2. % Answered $:= \mathbb{P}(|C| = 1)$; this measures the probability the agent outputs a single answer.

3. Specificity $:= 1 - \mathbb{E}[|C|/|\mathcal{Y}|]$; a higher specificity indicates that the model produces tighter prediction sets. This evaluates how well the model narrows down its prediction set (and uncertainty), reflecting its ability to focus on a smaller, more precise set of potential answers.

4. Efficiency $:= 1 - \mathbb{E}[t_{\text{ans}}/T]$; a more efficient agent has smaller stopping time, $t_{\text{ans}}$, i.e., the number of conversational turns required for the model to confidently produce a final prediction.

The probability and expectation are defined over the randomness in the distribution of the scenarios, the feedback and the evolution of the environment, and the internal randomness in the agent.

**Single-turn vs. multi-turn settings.** Traditional conformal prediction assumes a *single-turn* setting without interactions (Figure 1, left), where the goal is to construct prediction sets, $C(x)$, for fresh inputs, $x \in \mathcal{X}$, at test time satisfying the target coverage of $\mathbb{P}(y \in C(x)) \geq 1 - \alpha$. To this end, the goal of conformal prediction is to use the calibration set, denoted $D_{\text{cal}} = \{(x_i, y_i)\}_{i=1}^n$, and the model, denoted $\hat{f}$, to design a scheme that constructs small prediction sets at test time.

Our proposed CONFORMAL REASONING applies to a *multi-turn* setting (Figure 1, right), where the goal is to construct prediction sets, $C(\bar{x})$, about the hidden answer $y$ to a question for a fresh scenario $\xi = (e, y, T, x^0)$ at test time by interacting with the test scenario to get a sequence of

states $\bar{x} = (x^0, x^1, \ldots, x^{t_{\text{ans}}})$ for some $t_{\text{ans}} \leq T$. These sequence of interactions refine our belief about the answer, in order to construct a prediction set that satisfies the target coverage of $\mathbb{P}(y \in C(\bar{x})) \geq 1 - \alpha$. We interact with each scenario to get a calibration sequence, $\bar{x}_i = (x_i^0, x_i^1, \ldots)$, of evolving states. The goal of Conformal Reasoning is to design a scheme that guides the interactions to construct minimal size prediction sets using the scenarios in the calibration set.

## 4 BACKGROUND ON CONFORMAL PREDICTION

**Split Conformal Prediction (SCP) for single-turn settings.** In a non-interactive scenario, we are given a calibration set of paired examples, $D_{\text{cal}} = \{(x_i, y_i)\}_{i=1}^n \in \mathcal{X}^n \times \mathcal{Y}^n$, drawn i.i.d. and a model, $\hat{f}$, to predict $y$ given $x$. Given a new input $x$ and a target confidence level $1 - \alpha \in [0, 1]$, uncertainty quantification aims to produce a statistically valid prediction set, $C(x)$, of minimal size satisfying, $\mathbb{P}(y \in C(x)) \geq 1 - \alpha$, where the pair $(x, y)$ is drawn from the same distribution as $D_{\text{cal}}$.

To achieve this goal, SCP uses the model to define a *nonconformity score function, $s : \mathcal{X} \times \mathcal{Y} \to \mathbb{R}$.* This measures the disagreement between the prediction $\hat{f}(x)$ and the true label $y$. For example, $s(x, y) = \|y - \hat{f}(x)\|$ in regression and $s(x, y) = 1 - f_y(x)$ in classification, where $f_y(x)$ is the softmax score for class $y$ predicted from input $x$.

SCP calculates scores $S_{\text{cal}} := \{s_i = s(x_i, y_i)\}_{(x_i, y_i) \in D_{\text{cal}}}$ on the already-collected calibration dataset and computes the $(1 - \alpha)(1 + 1/n)$-th empirical quantile $\hat{q}$, i.e. $\hat{q} := Q_{(1-\alpha)(1+1/n)}(S_1, \ldots, S_n)$. Finally, the prediction set for a fresh input $x$ is the collection of outputs with small nonconformity scores defined as

$$C_\alpha(x) = \{\hat{y} \in \mathcal{Y} \,|\, s(x, \hat{y}) \leq \hat{q}_\alpha\},$$

and provably achieves the desired coverage guarantee; see Angelopoulos et al. (2023, Theorem D.1).

**Split Conformal Prediction (SCP) for multi-turn settings.** When considering split conformal prediction in an interactive environment, prediction sets must be constructed more delicately, especially at test time. First, recall from our interactive framework that each scenario, $\xi$, results in a sequence, $\bar{x} = (x^0, x^1, \ldots, x^{t_{\text{ans}}})$, for some $t_{\text{ans}} \leq T$; accordingly, since each scenario, $\xi$, is drawn i.i.d. from some distribution $\mathcal{D}$, this, along with the exploration policy, induces a distribution over input sequences $\bar{x}$. Let $\bar{x}_i$ denote the $i$-th trajectory drawn from this induced distribution for the scenario $\xi_i$. Ren et al. (2024) apply standard conformal prediction to these trajectories by defining nonconformity scores at each time step $t$ within a given trajectory as $s_i^t := s(x_i^t, y)$, where $x_i^t$ is the state that is fed into the model $\hat{f}$ at time $t + 1$.

As a baseline, we compare against a slight variation of the interactive SCP algorithm introduced in Ren et al. (2024). This SCP chooses the *episode-level* nonconformity score to be $\bar{s}_i := \max_{t \in [T]} s_i^t$, or the worst-case nonconformity score across all time steps in the trajectory. These *episode-level scores* can then be used for the standard approach in split conformal prediction; denote $\hat{q}_\alpha$ as the $1 - \alpha$ empirical quantile over $\bar{s}_i$. At test time, because the agent only observes the trajectory up to current time $t$, the prediction sets must be *causally* constructed. In particular, the prediction set at time $t$ is defined as

$$C(x^t) := \cap_{\tau=0}^t C(x^\tau), \text{ where } C(x^\tau) := \{\hat{y} \in \mathcal{Y} \,|\, s(x^\tau, \hat{y}) \leq \hat{q}_\alpha\}, \text{ for all } \tau \leq t,$$

for a fresh trajectory $(x^0, \ldots, x^t)$ on a test scenario $\xi$. If $|C(x^t)| = 1$, the agent halts exploration and returns the singleton set as its choice. If $|C(x^t)| = 0$ or $t = T$ is reached without a singleton prediction set, we need not return a prediction. This is in contrast with the version of SCP proposed in Ren et al. (2024), which output the most likely answer when $|C(x^t)| = 0$ or $t = T$; we make this change to emphasize our design choice of abstaining or requesting help from a human until accurate.

## 5 METHODS

While SCP attains marginal validity, it requires strong design choices that might not be desirable in general. We focus on three critical constraints of SCP—termination criteria, prediction set construction, and using the conformal sets in the policy—and propose CONFORMAL PREDICTION consisting of three novel conformal techniques—offline adaptation, refresh prediction, and guided reasoning—to address each of the constraints.

## 5.1 OFFLINE ADAPTATION (OA)

SCP requires the **termination criteria** of the calibration trajectories to be made *a priori*. Practically, existing work collect the calibration scores with a termination criterion (TC1) of a fixed trajectory length $T$ and select a quantile $\hat{q}$ to be the threshold for creating turn-level prediction sets at test time, but *vary the termination criterion at test time* (TC2) to be either reaching the max length $T$ *or* reaching a prediction set size of 1 dependent on the selected $\hat{q}$, causing a **distribution shift** in the score functions that is hard to avoid in practice[2](Ren et al., 2024). Given the strict (and often impractical) constraints under SCP in multi-turn interactive settings, we propose **offline adaptation** to address the distribution shift issue by dynamically updating the parameter $\alpha_t$ using a *training set* with the same score functions and data distribution as the test set. The learned $\alpha_t$ determines a new threshold $\hat{q}_{\alpha_t}$ from the calibration scores to provide **marginal coverage guarantees** on the test set.

Offline Adaptation is inspired by Adaptive Conformal Inference (Gibbs & Candes, 2021; Angelopoulos et al., 2024b, ACI), which addresses *online* uncertainty quantification in settings where arbitrary distribution shift might occur over time. In particular, at each time $t$, we consider a time-dependent bounded score function $s_t : \mathcal{X} \times \mathcal{Y} \to [0, B]$ defined similarly to before (see §4), but allowing for the score to evolve over time. Given $s_t$, the prediction set is defined as

$$C_t(x) = \{y \in \mathcal{Y} : s_t(x, y) \leq \hat{q}_{\alpha_t}\},$$

where $\hat{q}_{\alpha_t}$ is the $1 - \alpha_t$ empirical quantile over the collection of prior scores. The parameter $\alpha_t$ is updated according to

$$\alpha_{t+1} = \alpha_t + \eta_t(\alpha - \mathbb{1}_{y \notin C^t(x^t)}),$$

where $\eta_t \propto t^{-1/2-\epsilon}$ for some $\epsilon \in (0, 1/2)$. If $Y_t$ is not covered at time $t$, then $\alpha_t$ is increased, which in turn *decreases* the magnitude of $\hat{q}_{\alpha_{t+1}}$. A strength of ACI is its ability for $\alpha_t$ to converge almost surely to $\alpha^*$ when the data is i.i.d. but the score function is evolving over time. In particular, we appeal to the following theorem:

**Theorem 1** (Angelopoulos et al. (2024b)). Let $(X_t, Y_t) \overset{\text{i.i.d.}}{\sim} \mathcal{D}$ for some distribution $\mathcal{D}$, and assume the score functions $s_t$ are trained online. Assume that $\eta_t$ is a fixed nonnegative step size satisfying $\sum_{t=1}^{\infty} \eta_t = \infty$ and $\sum_{t=1}^{\infty} \eta_t^2 < \infty$. Let $s : \mathcal{X} \times \mathcal{Y} \to [0, B]$ be a fixed score function, and assume that $\alpha^*$ is unique as giving $1 - \alpha$ coverage of $s(X, Y)$. Then online conformal prediction satisfies the following statement almost surely: If $s_t \overset{\text{d}}{\to} s$, then $\alpha_t \to \alpha^*$. In other words, if the score functions converge in distribution to some fixed score function, $\alpha_t$ will converge to $\alpha^*$, where $\alpha^*$ corresponds to attaining $1 - \alpha$ coverage on the distribution of scores.

## 5.2 REFRESH PREDICTION (RP)

The **construction of the prediction sets** in SCP is hyperspecific: choosing a different trajectory-level score function or set construction method does not give statistical validity. SCP requires that a prediction set at time $t$ be constructed by the intersection of all previous prediction sets of time $j < t$; accordingly, if the true answer falls out of the prediction early in the calibration *or* inference trajectory, the set loses coverage no matter what. We emphasize that Theorem 1 holds for an *arbitrary* score function: as long as it converges in distribution, $\alpha_t$ will converge to $\alpha^*$. This enables conformal sets to be constructed flexibly using ACI, rather than through the restrictive and potentially conservative method required by SCP. Accordingly, we propose an alternative score function and method for constructing a conformal set. In particular, for a calibration trajectory $\bar{x} = (x^1, \dots, x^{\min(t_{\text{ans}}, T)})$, we define the score function as $s(\bar{x}, y) = s\left(x^{\min(t_{\text{ans}}, T)}, y\right)$, where $T$ is the maximum depth of the trajectory. I.e., only the nonconformity scores of each option at the *current step* are used to construct the prediction set, rather than using the intersection with all prior prediction sets. Thus, if an option is excluded from the prediction set in a previous step, it could still be included in a subsequent prediction set. We denote this approach for computing the prediction sets as *Refresh Prediction (RP)*, referring to the ability of the prediction set of being refreshed at each timestep.

---

[2]In an effort to unify the termination criteria for the calibration and test sets to eliminate the distribution shift, we explore the following two scenarios. Using TC1 for both calibration and test sets will result in the system being hypersensitive to the choice of $T$ *a priori*—small $T$ values result in insufficient exploration and large prediction sets, and large $T$ values result in inefficient and wasteful algorithms not customized to each environment—rending the set-up brittle and impractical. Using TC2 for both calibration and test sets will require the $\hat{q}$ value—needed to determine the turn-level prediction set—to be determined before iterating through the calibration set, causing a *circular dependency*.

---

**Algorithm 1:** Conformal Reasoning

---

**Input:** an exploration policy $\hat{\pi}(\cdot)$; a calibration set $D_{\text{cal}} = \{\xi_i\}_{i=1}^n$ of size $n$; initialization size $n_{\text{init}}$; a nominal error level $\alpha \in (0,1)$; and learning rate schedule with $\eta_j \propto j^{-1/2-\epsilon}$ for $\epsilon \in (0, 1/2)$, nonconformity score function $s(\cdot, \cdot)$

/* Collect an initial set of scores using $n_{\text{init}}$ sample sequences.    */

1 **for** $i \in [n_{\text{init}}]$ **do**
2   $\quad \bar{x}_i \leftarrow$ sequence of states from scenario $\xi_i = (e_i, T_i, x_i^0, y_i)$ run with policy $\hat{\pi}$
3   $\quad s_i = s(\bar{x}_i, y_i)$
4 $S := \{s_i\}_{i \in [n_{\text{init}}]}$
5 $\hat{q}_{\alpha_{n_{\text{init}}+1}} \leftarrow Q_{1-\alpha}\left(\delta_\infty + \sum_{s_i \in S} \delta_{s_i}\right)$
6 $\alpha_{n_{\text{init}}+1} \leftarrow \alpha$

/* Use $S$ to find $\hat{\alpha}$                                              */

7 **for** $j \in \{n_{\text{init}}+1, \ldots, n\}$ **do**
8   $\quad \bar{x}_j \leftarrow$ sequence of states from scenario $\xi_j = (e_j, T_j, x_j^0, y_j)$ run with policy $\hat{\pi}$
9   $\quad \alpha_{j+1} \leftarrow \alpha_j + \eta_j(\mathbb{1}_{y_j \notin C_j(\bar{x}_j)} - \alpha)$.
10  $\quad S \leftarrow S \cup \{s(\bar{x}_j, y_j)\}$
11  $\quad \hat{q}_{\alpha_{j+1}} \leftarrow Q_{1-\alpha_{j+1}}\left(\delta_\infty + \sum_{s_i \in S} \delta_{s_i}\right)$

**Output:** $\hat{q}_{\alpha_{n+1}}$

---

### 5.3 GUIDED REASONING (GR)

SCP prevents **using the conformal set**—i.e., the measure of the agent's uncertainty—in the decision-making policy; otherwise, the calibration set would also need to use conformal sets in its own decision-making policy, creating another circular dependency. Guided Reasoning enables the agent to use the conformal set in its *exploration* policy. In turn, this strategy aims to improve exploration quality by leveraging the model's uncertainty to guide subsequent actions. More formally, the agent takes action $a^t \sim \hat{\pi}(x^{t-1}, C(x^{t-1})$, where $C(x^{t-1})$ is the conformal set at time $t-1$. This creates an explicit dependence of the agent's action—and consequently, its subsequent state $x_t$—on the construction of the conformal set and therefore $\alpha_t$. Here, the use of OA is critical—because the calibration set was collected *without* using the conformal set in the exploration policy, it no longer provides marginal coverage over data collected with a policy that *does* leverage the conformal set. Notably, OA enable the model to adapt to the distribution shift while enabling use of the conformal set in the exploration procedure.

**Conformal Reasoning.** To this end, we introduce a novel approach, CONFORMAL REASONING, for maintaining coverage in interactive environments while leveraging conformal prediction sets in exploration; see Alg. 1 for further details. First, we partition a calibration dataset $D_{\text{cal}} = \{\xi_i | \xi_i \sim \mathcal{D}\}_{i=1}^n$ into two distinct subsets: $D_{\text{init}}$ and $D_{\text{train}}$. The subset $D_{\text{init}}$, where the policy runs until length $T$, is used for providing a warm start for OA. If the dataset is known ahead of time, this allows for additional efficiency through offline computation of $\alpha_0$ using traditional split conformal prediction. After $\alpha_0$ is initialized, we run adaptive conformal inference with a decreasing learning rate, where our update is only dependent on the *last* prediction set at $t_{\text{ans}}$. This allows us to use potentially non-causal methods for constructing our prediction set during exploration.

## 6 EXPERIMENTS

We compare the performance of our proposed techniques—**Offline Adaptation (OA)** with **Refresh Prediction (RP)** and **Guided Reasoning (GR)**, collectively termed CONFORMAL REASONING—to the baseline of Split Conformal Prediction (SCP) on the medical diagnosis (MediQ), embodied question answering (EQA), and 20 Questions (20Q) tasks described in §3. We defer more detailed description of the tasks to Appendix A. We experiment with target coverage values in $\{0.5, 0.6, 0.7, 0.8, 0.9\}$ to evaluate how varying coverage levels affect the key metrics for each task.

**Evaluation Metrics.** As described in §3, we measure the *Coverage*$:= \mathbb{P}(y \in C)$ of each method; *% Answered*—the rate at which the agent arrives at a single confident answer; *Specificity*—how well the model narrows down its prediction sets; and *Efficiency*—the speed at which the agent ends its exploration to produce a single confident answer.

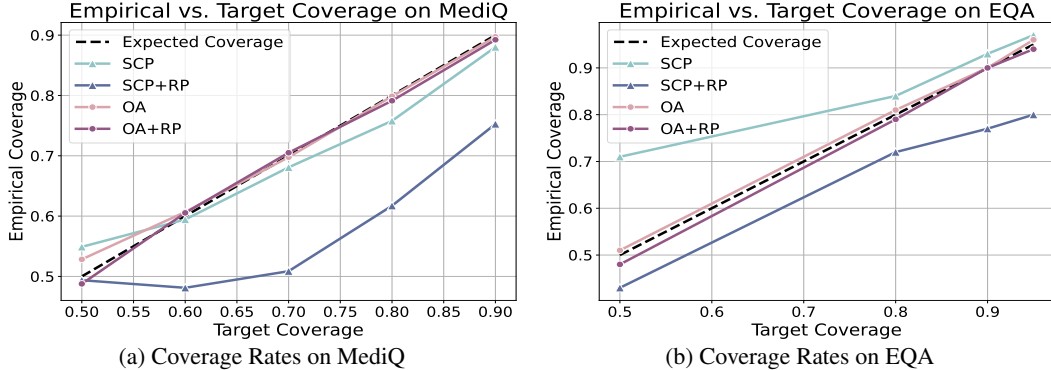

Figure 2: The proposed approaches of ACI and ACI+RP closely match the target coverage for both MediQ and EQA. The baseline of SCP and SCP+RP, on the other hand, fail to meet the target coverage. See Table 2 for exact values with standard errors.

**Models.** We rely on open models rather than proprietary black-box models, as we require probability outputs from our models. For MediQ and 20Q, we use the LLAMA-3-8B model (Dubey et al., 2024). For EQA, we use Prismatic VLM (Karamcheti et al., 2024), an open vision model that achieve strong performance on question-answering and spatial reasoning. See §A.2 for implementation details.

## 7    RESULTS

**Offline Adaptation (OA) maintains coverage.** A fundamental requirement of the conformal prediction framework is to ensure coverage, i.e., the prediction set consistently includes the correct answer with high enough probability. As expected, our Offline Adaptation technique ensures the coverage guarantee across all tasks by utilizing adaptive conformal inference, closely aligning empirical coverage with the target coverage level. In contrast, **SCP**, while theoretically sound in single-turn settings, struggles to maintain this guarantee in multi-turn settings (Figure 2). In particular, the draw of the calibration set and the corresponding conformal threshold influences the policies of the inference-time tasks, creating a distribution shift. In contrast, OA adjusts its prediction thresholds dynamically throughout the sample scenarios in the calibration set, ensuring that coverage remains close to the target level, even as the data distribution changes with the prediction thresholds. Upon further analysis, the deviation of SCP methods away from the target coverage is particularly pronounced when the trajectory lengths of the test set differ from that of the calibration set. More results on SCP across all three tasks can be found in Appendix B.

**Refresh Prediction (RP) increases percent answered and efficiency.** Refresh Prediction, a key part of our contribution, improves the model's ability to confidently select a single correct answer by the end of the interaction, which we measure using the *percent answered* metric. Using prediction sets constructed through RP, CONFORMAL REASONING is able to refine the model's final output, improving the frequency of the model outputting one single option by **42.7%** across tasks and target coverage levels (Table 2). Overall, RP improves the exploration efficiency by 20.7%, 20.9%, 55.1% on MediQ, EQA, and 20Q, respectively, effectively reducing the number of steps needed to achieve the target coverage while improving percent answered. This is particularly crucial in high-stakes or high-cost applications to ensure timely and resource-effective decisions. In medical diagnosis, this means that the model can confidently reach a conclusion after fewer questions, which is highly desirable in real-world applications where asking unnecessary questions can waste time or resources.

**Guided Reasoning (GR) improves accuracy for open-ended tasks.** Guided Reasoning enables the model to leverage its own uncertainty to produce the next best action. This is achieved by adding the prediction set from each turn to the prompt, guiding it to ask a question that would best distinguish the remaining options in the narrowed-down prediction set, rather than starting from the whole solution space. Intuitively, guided reasoning is the most helpful when the size of the solution space is large, as it helps the model focus on a subset of options only. On the 20Q task, GR not only improves efficiency and specificity, it significantly improves the accuracy by an average of 12.4% across target coverage levels with and without RP (figure 3). On the multiple-choice tasks, GR improves efficiency and specificity, though to a lesser degree than that of 20Q. Additional results on GR is reported in Appendix D.

| Task | Target Coverage | Refresh Prediction | Empirical Coverage | % Answered ↑ | Specificity(%)↑ | Efficiency(%)↑ |
|------|----------------|-------------------|-------------------|--------------|-----------------|----------------|
| MediQ | 0.9 | ✗ | $89.7_{\pm0.7}$ | $26.8_{\pm1.3}$ | $\mathbf{30.0}_{\pm1.1}$ | $22.9_{\pm1.0}$ |
| | | ✓ | $89.2_{\pm0.8}$ | $\mathbf{34.4}_{\pm1.2}$ | $27.9_{\pm0.9}$ | $\mathbf{28.9}_{\pm0.7}$ |
| | 0.8 | ✗ | $79.9_{\pm1.1}$ | $44.4_{\pm1.5}$ | $\mathbf{46.1}_{\pm1.4}$ | $38.1_{\pm1.5}$ |
| | | ✓ | $79.1_{\pm1.1}$ | $\mathbf{55.2}_{\pm2.5}$ | $45.0_{\pm1.7}$ | $\mathbf{47.7}_{\pm2.2}$ |
| | 0.5 | ✗ | $53.1_{\pm0.3}$ | $94.4_{\pm0.4}$ | $\mathbf{75.1}_{\pm0.0}$ | $91.1_{\pm0.1}$ |
| | | ✓ | $48.8_{\pm0.9}$ | $\mathbf{99.6}_{\pm0.5}$ | $74.9_{\pm0.1}$ | $\mathbf{96.7}_{\pm1.6}$ |
| EQA | 0.9 | ✗ | $90.3_{\pm0.6}$ | $13.1_{\pm1.2}$ | $30.5_{\pm0.8}$ | $14.4_{\pm3.7}$ |
| | | ✓ | $89.8_{\pm1.6}$ | $14.5_{\pm1.1}$ | $32.2_{\pm1.4}$ | $17.0_{\pm2.8}$ |
| | 0.8 | ✗ | $81.3_{\pm1.8}$ | $34.5_{\pm1.1}$ | $33.7_{\pm1.2}$ | $29.6_{\pm3.3}$ |
| | | ✓ | $79.7_{\pm1.4}$ | $\mathbf{42.9}_{\pm2.7}$ | $\mathbf{39.8}_{\pm1.5}$ | $\mathbf{42.2}_{\pm4.1}$ |
| | 0.5 | ✗ | $51.0_{\pm2.5}$ | $83.2_{\pm2.2}$ | $75.2_{\pm2.1}$ | $77.5_{\pm3.1}$ |
| | | ✓ | $48.1_{\pm2.3}$ | $83.7_{\pm1.8}$ | $\mathbf{84.0}_{\pm2.6}$ | $79.2_{\pm2.9}$ |
| 20Q | 0.9 | ✗ | $90.5_{\pm1.1}$ | $0.1_{\pm0.2}$ | $52.6_{\pm1.4}$ | $4.0_{\pm0.4}$ |
| | | ✓ | $88.4_{\pm2.0}$ | $\mathbf{5.7}_{\pm1.4}$ | $\mathbf{67.6}_{\pm1.4}$ | $\mathbf{7.9}_{\pm0.8}$ |
| | 0.8 | ✗ | $83.2_{\pm3.1}$ | $0.5_{\pm0.4}$ | $62.4_{\pm2.1}$ | $4.6_{\pm0.5}$ |
| | | ✓ | $82.7_{\pm4.6}$ | $\mathbf{10.8}_{\pm2.1}$ | $\mathbf{77.5}_{\pm2.5}$ | $\mathbf{8.9}_{\pm1.1}$ |
| | 0.5 | ✗ | $53.4_{\pm3.8}$ | $2.5_{\pm0.5}$ | $79.5_{\pm0.4}$ | $4.8_{\pm0.5}$ |
| | | ✓ | $52.5_{\pm3.6}$ | $\mathbf{38.9}_{\pm4.5}$ | $\mathbf{93.7}_{\pm0.2}$ | $\mathbf{14.3}_{\pm1.2}$ |

Table 2: Conformal Reasoning results on MediQ, EQA, and 20Q, with depth limit 15, 30, and 20 respectively. Conformal Reasoning employs adaptive conformal prediction to ensure that empirical coverage matches the target coverage, while adding refresh prediction (RP) further improves percent answered and exploration efficiency without significantly sacrificing output specificity.

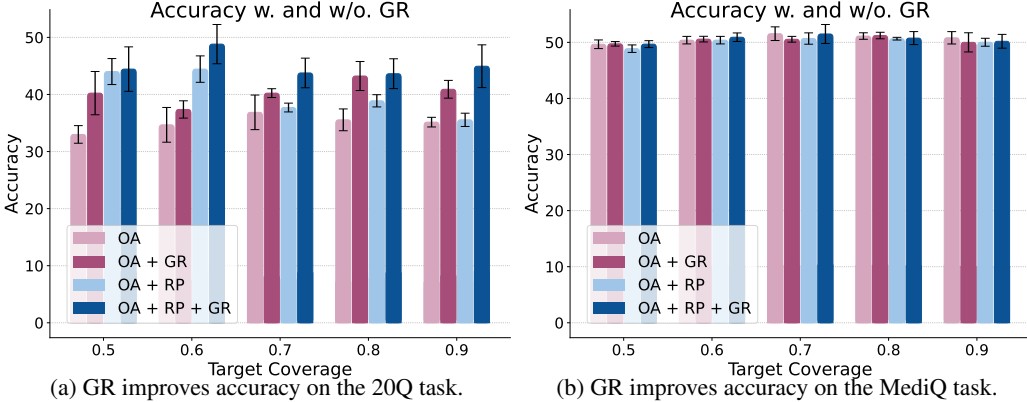

(a) GR improves accuracy on the 20Q task.     (b) GR improves accuracy on the MediQ task.

Figure 3: Guided Reasoning (GR) significantly improves accuracy on tasks with a larger solution space (20Q) while maintaining performance on others. More results in Appendix D.

## 8 DISCUSSION

In this work, we introduce **CONFORMAL REASONING**—extending adaptive conformal inference to interactive environments. We first propose a novel offline adapataion paradigm where the need for sample-wise feedback is eliminated, allowing practical deployment in real-world scenarios. We present 1) **Refresh Prediction** for constructing flexible prediction sets in multi-turn settings and 2) **Guided Reasoning** for leveraging the prediction set in the decision-making process. Our results demonstrate strong empirical performance in domains like medical diagnosis, embodied question answering, and open-ended 20 questions, showing more reliable coverage rate and improved exploration efficiency. However, there are still various important considerations to explore in the space of conformal prediction for multi-turn interactions. While Conformal Reasoning addresses the heuristic bias between calibration trajectories and termination criteria, we encourage future research to focus on optimizing reasoning abilities of agents with episode-level prediction sets and understanding trade-offs among coverage, efficiency, and specificity.

LIMITATIONS

While Conformal Reasoning maintains coverage and improves performance relative to split conformal prediction, our approach nonetheless has some limitations. First, convergence and stability of Conformal Reasoning is dependent on the choice of learning rate in adapting $\alpha_t$, requiring some hyperparameter tuning for the best performance. Moreover, if the calibration set for tuning alpha is too small, Conformal Reasoning might not converge to the optimal $\alpha^*$ value.

ETHICS STATEMENT

While the Conformal Reasoning framework shows potential, several risks must be addressed before real-world deployment.

**Bias and Fairness.** Calibration sets can introduce biases, which may result in unequal outcomes across different demographics, especially in medical applications. Future work should assess these risks through fairness evaluations to ensure equitable outcomes across all groups.

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

# A  DETAILED TASK DESCRIPTIONS

## A.1  MEDIQ

We now include details about how the agent performs clinical reasoning using the MediQ framework (Li et al., 2024). The MediQ framework comprises two components: a *Patient system* that simulates a patient and responds to follow-up questions, and an *Expert system* that serves as a doctor's assistant and asks questions to the patient before making a medical decision. We MediQ to explore how conformal prediction can guide the Expert System's interactive medical diagnosis capabilities. Specifically, in this **interactive clinical reasoning task**, a successful information-seeking Expert should decide, at each turn, whether it has enough information to provide a confident answer; if not, it should ask a follow-up question.

The dynamic medical consultation task simulates the iterative nature of real-world clinical interactions. This task starts by providing an initial patient description $k_0$ of their conditions to the Expert system. The initial information typically contains the patient's age, gender, and chief complaint for the visit. The Patient system has access to the entire patient record $\mathcal{K} = \{k_0, k_1, \ldots, k_n\}$, and the necessary information to answer the multiple choice question is $\mathcal{K}^* \subseteq \mathcal{K}$. At the start of the $t$-th turn, the knowledge available to the Expert system is denoted as $\mathcal{K}_{t-1} = \{k_0, \ldots, k_i\}$. Given follow-up question $q_t$, the Patient system responds with $r_t = \{k | k \in \mathcal{K}\}$. The Expert knowledge is then updated as $\mathcal{K}_t = \mathcal{K}_{t-1} \cup r_t$. The **main challenge of the task** is for the Expert system to ask information-seeking questions to expand $\mathcal{K}_t$ until the Expert system is sufficiently, at which point the Expert system is asked to make a final decision. The decision to either ask another follow-up question or to provide the final diagnosis is made through the episode-level prediction set generated by the conformal prediction process $(C(x^t))$—when there are more than one option remaining in the prediction set, more information is needed to rule out the wrong answers, so the model should abstain from answering, whereas when there is only one option in the prediction set, we deem the model confident enough to provide the final answer.

**Nonconformity Scores** At each step of the interaction, the model outputs logit scores for the possible letter choices—A, B, C, D—which are conditioned on the preceding conversation context and the given question. We apply a softmax function to these logits to derive normalized probabilities, which we use as nonconformity scores for each option.

## A.2  EQA

Here, we provide additional details on *how* exploration and question-answering takes place in our embodied question answering tasks. We focus on two parts in particular: how EQA performs targeted exploration and the scores used as the stopping criterion in the conformal prediction subroutine. We closely follow the approach described by Ren et al. (2024); we restate this approach here briefly for further detail but refer the reader in search of full implementation details to ren2024exploreconfident.

### A.2.1 TARGETED EXPLORATION

In EQA, we would like the robot to explore areas that are useful in answering the desired question. To do so, we create a 3D voxel representation for the scene, where each voxel corresponds to a cube of fixed side length. At a given state of the robot—that is, a pose and location within the environment—and associated depth image, we apply volumetric truncated signed distance function (TSDF) to update the occupancy of the voxels and their presence in the current image. At each time step, the 3D voxel map is projected onto a 2D map $M$, where each 2D point is determined as occupied or unoccupied based on the occupancy of voxels within a 1.5m radius and as explored or not if the voxels have been marked as explored. Based on the constructed 2D map, we then use a heuristic-based 2D planner based on Frontier-Based Exploration (FBE). This approach looks for locations at the boundary of explored and unexplored areas, samples one, and uses the normal vector from the current location to the selected region boundary as the target orientation.

While naive frontier-based exploration can yield efficient exploration of an environment, it fails to leverage the prior knowledge stored within the VLM for determining the most relevant areas for exploration. Here, we characterize the VLM's uncertainty using visual prompting. For a given image, we identify three points in free space within the image and annotate them on the image with the letters `A`, `B`, and `C`. We then use this image for visual prompting using the prompts identified in Figure 4. The normalized output probability can be used to construct the *local semantic value* of each of the three points as follows:

$$LSV_p(x^t) = \hat{f}_{y_p}(x^t) = \hat{f}_{y_p}(I_c^t, s_{\text{LSV,q}}) \in [0, 1]$$

where $I_c^t$ is the current RGB image, $q$ is the question to be answered, $\hat{f}_y(\cdot)$ is the (normalized) softmax score associated with token $y$, and $s_{\text{LSV,q}}$ is the prompt from Figure 4 with the question filled in. In particular, $\hat{f}_{y_p}(\cdot)$ is the softmax score associated with the letter for a given point $p$, i.e. if $p$ is annotated with the letter `A`, then $\hat{f}_{y_p}(\cdot)$ is the normalized softmax score associated with the token `A`.

While LSV captures the local score associated with each point, it does not capture the value of different points seen in alternate poses of the robot. To help address this problem, we determine whether we should navigate to other poses in the first place. We can again prompt the VLM using prompt in Figure 5. This captures the global semantic value of moving to point $p$, and is formulated as

$$GSV_p(x^t) = \hat{f}_{\text{Yes}}(x^t) = \hat{f}_{\text{Yes}}(I_c^t, s_{\text{GSV,q}}) \in [0, 1]$$

In other words, we characterize the the value in moving to point $p$ by the normalized softmax score of the VLM predicting "yes." Together, we compute the overall semantic value as

$$SV_p(x^t) = \exp(\tau_{\text{LSV}} \cdot LSV_p(x^t) + \tau_{\text{GSV}} \cdot GSV_p(x^t))$$

where $\tau_{\text{LSV}}$ and $\tau_{\text{GSV}}$ are temperature-scaling parameters. We then use this semantic value map weighting our frontier-based exploration value map.

### A.2.2 NONCONFORMITY SCORES

Here, we also characterize the nonconformity scores used in conformal reasoning. First, we can characterize the VLM's confidence in the current image being relevant for answering the target question through direct prompting. In particular, we define the the question-image relevance score as

$$Rel(x^t) = \hat{f}_{\text{Yes}}(I_c^t, (q, s_{\text{Rel, q}}))$$

where $s_{\text{Rel,q}}$ is the prompt in Figure 6 with the question filled in. Finally, we determine the relevance-weighted confidence score at time $t$ as

$$\rho_y^t(x^t) := Rel(x^t)(\hat{f}_y(x^t) - 1).$$

We use this as our nonconformity score for our EQA task.

### A.3 IMPLEMENTATION DETAILS

At each step of the interaction, the model outputs logit scores for the possible choices—`A,B,C,D` for multiple choice and `yes,no` for open-ended questions—which are conditioned on the preceding

```
Consider the question: {question}, and you will explore the scene for
answering it.  Which direction (black letters on the image) would you
explore then?  Answer with a single letter.
```

Figure 4: EQA local semantic value (LSV) prompt.

```
Consider the question: {question}, and you will explore the scene for
answering it.  Is there any direction shown in the image worth exploring?
Answer with Yes or No.
```

Figure 5: EQA global semantic value (GSV) prompt.

```
Consider the question {question}.  Are you confident about answering the
question given the current view?
```

Figure 6: EQA relevance score prompt.

conversation context and the given question. We apply a softmax function to these logits to derive normalized probabilities, which we use as nonconformity scores for each option. In the open-ended setting, we use a preset list of 26 options consisting of the correct answer and in-category and out-of-category terms to simulate a large solution space. In our EQA task, we weight the normalized probabilities by a question-image relevance score; see Appendix A for task-specific details.

## B    SCP RESULTS

| Task | Method | Avg. Deviation | Avg. Efficiency |
|------|--------|----------------|-----------------|
| MediQ | **SCP+X** | $5.70_{\pm 5.33}$ | $62.07_{\pm 23.98}$ |
|       | **OA+X** | $1.38_{\pm 1.09}$ | $53.86_{\pm 24.35}$ |
| EQA | **SCP+X** | $90.2_{\pm 0.7}$ | $21.4_{\pm 2.0}$ |
|     | **OA+X** | $89.2_{\pm 0.8}$ | $\mathbf{31.0}_{\pm 1.8}$ |
| 20Q | **SCP+X** | $3.50_{\pm 2.72}$ | $7.43_{\pm 3.14}$ |
|     | **OA+X** | $3.42_{\pm 2.86}$ | $7.30_{\pm 3.13}$ |

Table 3: SCP on multi-turn interactive tasks weakens coverage guarantees.

## C    ABLATIONS

**Depth Limit**    In this ablation study, we control for the lengths of the trajectories to highlight the heuristic bias introduced by the calibration set and train-test sets not having the same termination criteria. By shortening the maximum number of turns in each trajectory, we reduce the variations between the lengths in the calibration set and the train-set sets. As shown in Table 5, after increasing the maximum depth of the trajectories, the distribution, both in mean and standard error, changes relative to trajectories with shorter depth limit. These differences are particularly noticeable with respect to efficiency.

**Online Conformal Algorithm**    While ACI is the most basic form of base algorithm to use for offline adaptation (OA), more modern approaches to online conformal inference exist, particularly with respect to controlling different measures of regret. Here, we compare our results against two more recent online conformal techniques: MVP from Bastani et al. (2022) and MagL-D from Zhang et al.. Our results on the EQA are available in Table 4. Broadly, we demonstrate that our approach is agnostic of the specific online conformal algorithm used; that is, our approach still maintains coverage across our design choices. Moreover, while using the improved online conformal algorithms

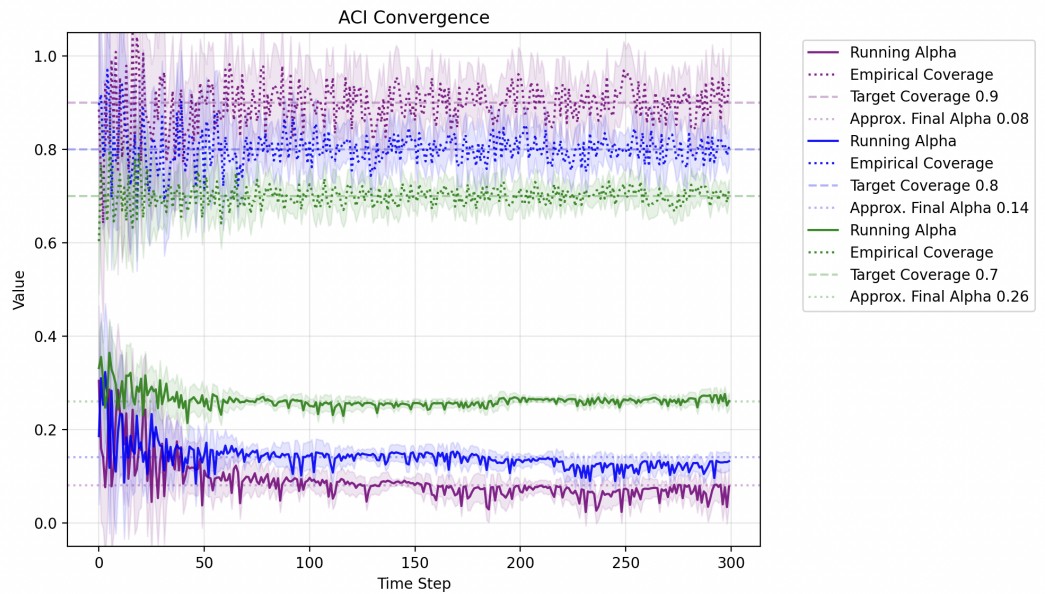

Figure 7: Empirical OA convergence on EQA task across coverage levels. Despite some variation in $\alpha_t$ and empirical coverage level, OA tends to converge relatively quickly despite its guarantees primarily being asymptotic.

improves our downstream metrics, they do so relatively marginally compared to ACI, which is simpler to implement. Characterizing the tradeoffs in downstream metrics between these methods is interesting future work.

**Empirical Convergence of** $\alpha_t$    While the theoretical guarantees for using offline adaptation come asymptotically, we characterize the empirical convergence of $\alpha_t$ for our EQA task in Figure 7. Here, we see that across coverage levels, OA empirically converges reasonably quickly despite being applied on a relatively small calibration set size.

**Tradeoffs Between Efficiency and Specificity in Refresh Prediction**    Here, we characterize the tradeoffs between specificity and efficiency, particularly in the context of RP. Our results are visible in Figure 8. Broadly speaking, we see that in many cases, using RP improves the Pareto frontier relative to OA alone. Additionally, the degree to which the Pareto frontier changes under RP strongly depends on the task that Conformal Reasoning is applied to.

## D    GUIDED REASONING RESULTS

Experimental results find that Guided Reasoning (GR) is particularly helpful when the size of the solution space is large. However, we also show that for multiple choice tasks containing only 4 options, addiing GR still maintains robust performance on both the MediQ (Figure 3b) and EQA (Figure 9) tasks.

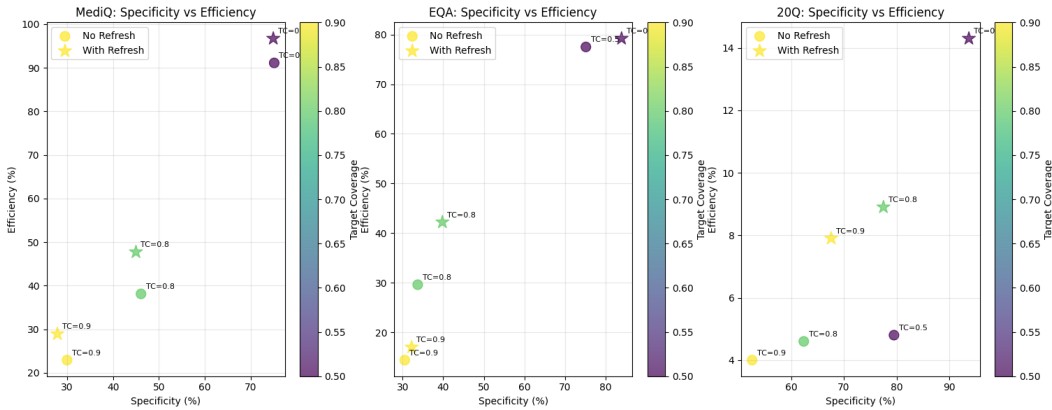

Figure 8: Tradeoffs between specificity and efficiency across tasks and target coverage levels. In general, using RP improves the Pareto frontier between specificity and efficiency.

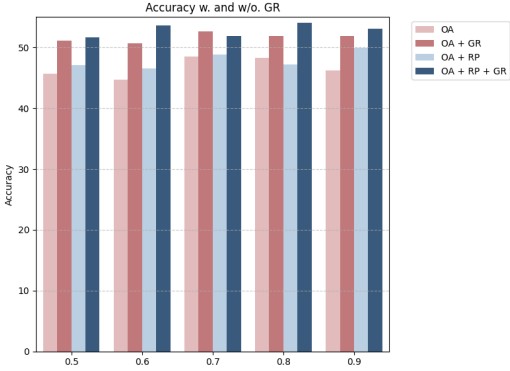

Figure 9: GR Performance on EQA. Relative to OA and OA+RP, adding GR improves accuracy on EQA.

| OA Alg. | Target Coverage | Refresh Prediction | Empirical Coverage | % Answered ↑ | Specificity(%)↑ | Efficiency(%)↑ |
|---|---|---|---|---|---|---|
| ACI | 0.9 | ✗ | $90.3_{\pm0.6}$ | $13.1_{\pm1.2}$ | $30.5_{\pm0.8}$ | $14.4_{\pm3.7}$ |
|  |  | ✓ | $89.8_{\pm1.6}$ | $14.5_{\pm1.1}$ | $32.2_{\pm1.4}$ | $17.0_{\pm2.8}$ |
|  | 0.8 | ✗ | $81.3_{\pm1.8}$ | $34.5_{\pm1.1}$ | $33.7_{\pm1.2}$ | $29.6_{\pm3.3}$ |
|  |  | ✓ | $79.7_{\pm1.4}$ | $\mathbf{42.9}_{\pm2.7}$ | $\mathbf{39.8}_{\pm1.5}$ | $\mathbf{42.2}_{\pm4.1}$ |
| MVP | 0.9 | ✗ | $90.1_{\pm0.7}$ | $14.2_{\pm1.1}$ | $32.1_{\pm0.9}$ | $15.8_{\pm3.5}$ |
|  |  | ✓ | $89.5_{\pm1.5}$ | $15.8_{\pm1.2}$ | $33.9_{\pm1.3}$ | $18.5_{\pm2.9}$ |
|  | 0.8 | ✗ | $81.0_{\pm1.7}$ | $36.2_{\pm1.2}$ | $35.4_{\pm1.3}$ | $31.8_{\pm3.4}$ |
|  |  | ✓ | $79.4_{\pm1.5}$ | $\mathbf{44.5}_{\pm2.5}$ | $\mathbf{41.6}_{\pm1.6}$ | $\mathbf{44.7}_{\pm4.0}$ |
| MagL-D | 0.9 | ✗ | $90.2_{\pm1.0}$ | $14.8_{\pm1.3}$ | $33.2_{\pm1.1}$ | $16.5_{\pm3.6}$ |
|  |  | ✓ | $89.3_{\pm1.7}$ | $16.4_{\pm1.2}$ | $34.8_{\pm1.4}$ | $19.2_{\pm2.7}$ |
|  | 0.8 | ✗ | $80.8_{\pm1.9}$ | $37.8_{\pm1.3}$ | $36.9_{\pm1.4}$ | $33.5_{\pm3.5}$ |
|  |  | ✓ | $79.1_{\pm1.6}$ | $\mathbf{46.2}_{\pm2.4}$ | $\mathbf{43.3}_{\pm1.7}$ | $\mathbf{46.8}_{\pm3.9}$ |

Table 4: Conformal Reasoning results on EQA using different online conformal algorithms. Broadly speaking, we see that using more advanced forms of online conformal algorithms yield benefits across our downstream metrics while retaining coverage.

| Exp. | Target Coverage | Max Depth | Empirical Coverage | % Answered ↑ | Specificity(%)↑ | Efficiency(%)↑ |
|---|---|---|---|---|---|---|
| OA | 0.9 | 10 | $89.3_{\pm0.7}$ | $17.2_{\pm0.9}$ | $29.2_{\pm1.2}$ | $09.6_{\pm1.1}$ |
| OA+RP | 0.9 | 10 | $89.5_{\pm1.4}$ | $18.4_{\pm1.3}$ | $28.9_{\pm1.7}$ | $12.1_{\pm0.8}$ |
| OA | 0.9 | 30 | $90.3_{\pm0.6}$ | $13.1_{\pm1.2}$ | $30.5_{\pm0.8}$ | $14.4_{\pm3.7}$ |
| OA+RP | 0.9 | 30 | $89.8_{\pm1.6}$ | $14.5_{\pm1.1}$ | $32.2_{\pm1.4}$ | $17.0_{\pm2.8}$ |
| OA | 0.8 | 10 | $80.2_{\pm1.0}$ | $24.4_{\pm1.1}$ | $32.2_{\pm1.5}$ | $15.9_{\pm1.8}$ |
| OA+RP | 0.8 | 10 | $80.9_{\pm0.9}$ | $24.1_{\pm1.6}$ | $32.1_{\pm1.7}$ | $16.1_{\pm1.4}$ |
| OA | 0.8 | 30 | $81.3_{\pm1.8}$ | $34.5_{\pm1.1}$ | $33.7_{\pm1.2}$ | $29.6_{\pm3.3}$ |
| OA+RP | 0.8 | 30 | $79.7_{\pm1.4}$ | $42.9_{\pm2.7}$ | $39.8_{\pm1.5}$ | $42.2_{\pm4.1}$ |
| OA | 0.5 | 10 | $43.9_{\pm8.3}$ | $81.3_{\pm2.6}$ | $84.0_{\pm1.2}$ | $74.2_{\pm1.5}$ |
| OA+RP | 0.5 | 10 | $45.7_{\pm7.2}$ | $85.1_{\pm2.1}$ | $80.2_{\pm2.9}$ | $75.4_{\pm1.1}$ |
| OA | 0.5 | 30 | $51.0_{\pm2.5}$ | $83.2_{\pm2.2}$ | $75.2_{\pm2.1}$ | $77.5_{\pm3.1}$ |
| OA+RP | 0.5 | 30 | $48.1_{\pm2.3}$ | $83..7_{\pm1.8}$ | $84.0_{\pm2.6}$ | $79.2_{\pm2.9}$ |

Table 5: Ablation study on depth limit on the EQA task. While coverage remains essentially the same, changing the depth limit in EQA affects other downstream metrics, particularly efficiency.

