# OpenReview forum: "Conformal Reasoning: Uncertainty Estimation in Interactive Environments"
_ICLR.cc/2025/Conference — Submitted to ICLR 2025_

### Official Review · Reviewer_WKSU · 2024-10-29

**Soundness:** 3
**Presentation:** 3
**Contribution:** 2
**Rating:** 6
**Confidence:** 2

**Summary:**

This paper studies conformal prediction in a multi-turn setting, where an agent interacts with a user repeatedly, adjusting its predictions at each turn. The authors first identify two major limitations in traditional split conformal prediction (SCP): (1) SCP requires the prediction set at time $t$ to be the intersection of all previous sets, which can be overly restrictive, and (2) SCP calibrates a quantile using offline data specific to a certain policy and cutoff parameter, which may not generalize well out-of-sample due to distributional shifts. To address these challenges, the authors employ adaptive conformal prediction (ACP), which dynamically adjusts the quantile value as the environment changes and relaxes the restrictive assumptions in (1). The main contribution of this paper is the application of ACP in this multi-turn context, where it demonstrates better empirical performance over SCP in the authors' experiments.

**Strengths:**

- Overall, the paper is well-written. The concepts and split and adaptive conformal prediction are clearly explained.

- Conformal prediction in this multi-turn setting is an important problem that has application in many settings (especially those that involves LLM).

- Computational experiments demonstrate favorable performance of the proposed method.

**Weaknesses:**

- **Motivation.** I am not sure if I fully understand why the traditional SCP requires the assumption that the current prediction set should belong to the intersection of previous ones. This seems to be a design choice that can be avoided. I suggest the authors providing more contexts regarding why this is necessary to better motivate the application of ACP.

- **Experiments.** For an applied paper like this, I suggest the authors consider the following points to strengthen their empirical contributions.

  - **Coverage**. It's unclear to me how the coverage metric was calculated. Specifically, did you calculate the coverage for every time step or only do this at the final step?
  - **Domain shift.** One of the motivations for applying ACP is to deal with domain shifts. However, this is not tested in the computational experiments.
  - **RP.** It is unclear to me why RP helps to improve \% answered, although the experiment results said so. It would be helpful if the authors could provide high-level intuition on the reason.
  - **Efficiency.** It is unclear what the baselines are when the authors claim that conformal reasoning improves efficiency.

**Questions:**

By breaking the connection between prediction sets constructed at consecutive steps, ACP and RP enhance flexibility in generating prediction sets. However, this approach might compromise interpretability. It would be interesting to know if the authors have considered this trade-off and have any insights on it. Additionally, it could be valuable to explore potential approaches that lie between these two extremes.

---

> ### Author Response · Authors · 2024-11-24
>
> Thank you for your thoughtful review. We appreciate your acknowledgment of our strengths, including the well-written prose, the importance of the problem, and strong computational experiments. We address your concerns below, as well as in the revised manuscript.
>
> 1. **Motivation of traditional SCP set construction.**
>
> > why the traditional SCP requires the assumption that the current prediction set should belong to the intersection of previous ones.
>
> For a given test sequence $(x^1, \dots, x^t)$ through time $t$, Ren et al. (2024) aim to use split conformal prediction to construct a conformal set that uses the information in the entire sequence of a scenario, $(x^0, \dots, x^T)$, to determine the non-conformity score of the test scenario.
> To do so, they define a score corresponding to a *single* timestep as $s_y^t(x^t)$; this is similar to a typical non-conformity score used in single-timestep conformal prediction.
> Subsequently, they define the non-conformity score of the *entire sequence* $(x^1, \dots, x^T)$ as the maximum non-conformity score across all timesteps in the sequence, or
> $$ s_y(x^1, \dots, x^T) = \max_{1\leq t \leq T} s_y^t(x^t). $$
> In other words, this is a worst-case bound on the non-conformity score of the test scenario.
> As a result, they create a sequence-level prediction set that attains $1-\alpha$ marginal coverage across the dataset by using the quantile of these worst-case non-conformity scores $\hat{q}\_{\alpha}$.
> However, this score is technically defined as a function of the entire sequence; at test time, however, the agent only has access to the sequence up to the current timestep.
> To render this score applicable at *test time*, using the previously defined quantile $\hat{q}_{\alpha}$, the prediction set at time $t$ is given by
>
> $$C_t(x^t) = \lbrace y \in \mathcal{Y} : s_y(x^1, \dots, x^t) \leq \hat{q}_{\alpha}\rbrace.$$
>
> In turn, they demonstrate that for all time $t\in [T]$ the sequence-level prediction set will be contained in $C_t(x^t)$, and in fact that the sequence-level prediction set can be written as the intersection of the prediction sets at each timestep, or
> $$ C(x^1, \dots, x^T) = \bigcap_{t=1}^T C_t(x^t). $$
> Accordingly, this worst-case construction helps ensure that the sequence-level prediction set---which with high probability contains the true label $y$---will be a subset of the prediction set at any single timestep.
> However, for the reasons we articulate below (Point 4), this worst-case construction may be too conservative.
> Additionally, we provide empirical results that SCP with other set construction methods fail in Figure 2 of the paper.
> We will take care to clarify this point in the revised manuscript.
>
> 2. **Calculation of Coverage**
> > It's unclear to me how the coverage metric was calculated. Specifically, did you calculate the coverage for every time step or only do this at the final step?
>
> Our coverage metric is calculated as follows: for each test point, we write that the point has been covered by the prediction set if the **final step** prediction set contains the true label.
> More formally, $\text{Coverage} = \frac{1}{n} \sum_{i=1}^n \boldsymbol{1}[y_i \in C_T(x_i^T)]$, where $T$ is the final prediction step.
> We will make this explicit in the revised manuscript.
>
> 3. **Domain shift.**
>
> > One of the motivations for applying ACP is to deal with domain shifts. However, this is not tested in the computational experiments.
>
> While ACI was initially introduced for the explicit purpose of adapting to distribution shifts, we note that in subsequent work, adaptive conformal-style approaches have also been characterized for online i.i.d. data.
> However, in practice, much of the data collected for interactive tasks is non-i.i.d., even when the agent's policy is stationary (or fixed a priori).
> In particular, implicit heuristics in how *calibration* data is collected in interactive settings---for example, the maximum sequence length, the agent's stopping threshold---can lead to non-i.i.d. data and accordingly non-exchangeable non-conformity scores, much less data coming different distributions (e.g. heterogeneous types of medical questions or household navigation scenarios).
> Part of the intuition in our approach is to correct for non-i.i.d data by leveraging the ability to adapt $\alpha_t$ over the potentially heterogeneous calibration set.
> Accordingly, even though the data used for calibration is non-i.i.d., adaptive conformal methods can still empirically provide coverage guarantees, even when we keep the learned $\alpha$ fixed when deploying at test time.
> We make this underlying motivation more explicit in the revised manuscript.

---

> ### Author Response · Authors · 2024-11-24
>
> 4. **Refresh prediction and answer rate.**
>
> The general intuition around refresh prediction is that it acts as a flexible method for constructing prediction sets that are more responsive to the test sequence.
> For example, consider the following scenario:
>
> In the EQA task, an autonomous agent is tasked with answering a question about its environment with four possible answers; WLOG, assume the correct answer is $y_1$.
>
> If the intersection-based method for constructing prediction sets is used, at an early timestep, the agent may produce a prediction set that does not contain the correct answer, e.g., $\{y_2, y_3, y_4\}$.
> However, after further interaction with the environment, even if the agent's prediction set at a later timestep is exclusively the correct label $\{y_1\}$, the final prediction set will not contain $y_1$, since the conformal set at time $t$ is defined as $C(x^t)=\cap_{i=1}^t C_i(x^i)$ at test time.
> In contrast, refresh prediction considers only the prediction set at each timestep; accordingly, the agent can "rediscover" the correct answer $y_1$ at later timesteps by "refreshing" its prediction set.
>
> Moreover, using the intersection-based method for constructing prediction sets defines its non-conformity score as the maximum (worst-case) non-conformity score across all timesteps, which may be too conservative.
> In contrast, refresh prediction's non-conformity score is the non-conformity score at the final timestep in the sequence, whether the result of encountering the maximum sequence length or by choosing a specific label.
> As a result, refresh prediction tends to be *less conservative* than the intersection-based method, particularly when the agent encounters information that is indicative of the correct label.
>
> 5. **Efficiency / metric benchmarks.**
>
> > It is unclear what the baselines are when the authors claim that conformal reasoning improves efficiency.
>
> In our experiments, we compare our approach against split conformal prediction (SCP); additionally, when ablating different components of our approach (e.g. RP), we similarly compare against SCP with RP.
> Moreover, each of our metrics mentioned in our manuscript is applicable to any prediction set-based method used in our interactive problem settings.

---

### Official Review · Reviewer_8aEW · 2024-11-01

**Soundness:** 3
**Presentation:** 3
**Contribution:** 2
**Rating:** 5
**Confidence:** 2

**Summary:**

This paper considers the conformal reasoning problem in a multi-turn environment. The authors introduce the conformal reasoning problem and focus on the confidence set construction for this problem. They propose the adaptive conformal inference and refresh prediction as two main components. The efficacy of the proposed algorithm is verified via experimental results.

**Strengths:**

1.	The paper introduces the problem formulate and the algorithm clearly.
2.	The authors provide theoretical analysis of the proposed method in Theorem 1.

**Weaknesses:**

1.	The logic behind Section 4.1 is not clear. The authors claim that ``While much of the traditional conformal prediction literature assumes i.i.d. (or exchangeable) sampled data or a calibration set collected offline’’. However, the authors adopt the i.i.d. assumption in Theorem 1. If the non-stationary factors can be absorbed in $s_t$, then the convergence of $s_t$ to $s$ will be an unrealistic assumption in the setting considered by this work.
2.	Theorem 1 is a result from the existing works where the setting is different from the proposed multi-turn task. It is encouraged to derive the analysis of the algorithm for the setting proposed in this work.
3.	Table 2 indicates that refresh prediction indeed sacrifices the specificity for the improvement of efficiency. It is beneficial to simulate and discuss the pareto frontier of the trade-off between these two factors.

**Questions:**

Same as the weakness

---

> ### Author Response · Authors · 2024-11-24
>
> Thank you for your helpful response. We aim to answer your questions in the comments below in addition to the revised manuscript.
>
> > Logic behind Section 4.1 is not clear
>
> We welcome your question on the intuition around Section 4.1, and we agree that the discussion could be more clear.
> Here, we hope to characterize the expected behavior of $\alpha_t$ over the course of the interaction.
> In the i.i.d. setting, we expect $\alpha_t$ to converge as the agent interacts with the environment.
> While our setting retains that same assumption---that is, that the scenarios sampled are i.i.d. from the distribution over our scenarios---the trajectories created under an agent's exploratory policy can appear to be heterogenous.
> More concretely, although scenarios $\xi$ and $\xi'$ are sampled from the same distribution, the trajectories $\bar x_\xi = (x^0,\dots, x^T)$ and $\bar x_\xi' = (x'^0,\dots, x'^T)$ may be quite different because of the randomness in the agent's policy and differences in the scenario, such as the target question.
> An analogous way of viewing this is with respect to time series: while two time series $\{X_t\}$ and $\{X_t'\}$ may be sampled from the same distribution, their trajectories may have significant differences.
> As a result, using offline adaptation can better address this heterogeneity in the trajectories.
>
> > Theorem 1 is a result from the existing works where the setting is different from the proposed multi-turn task. It is encouraged to derive the analysis of the algorithm for the setting proposed in this work.
>
> We argue that the setting used in Theorem 1---that of an online setting---is nonetheless a natural setting to characterize, even for our approach.
> In Angelopoulos et al. (2024), the authors prove a general result across arbitrary distributions of covariate label pairs that arrive in an online fashion.
> In our setting, we treat our covariates as the sequences of states of an agent interacting with its environment, with the corresponding label as the ground truth answer desired by the agent.
> Because of this construction, we can nonetheless apply the same results from adaptive conformal inference to our setting.
> We clarify this point in the revised manuscript.
>
> Moreover, even in the non-i.i.d. setting, we reference the result from Gibbs and Candes (2021) that characterizes finite-sample convergence of the adaptive procedure under Markovian data; accordingly, even in settings where the data distribution explicitly changes relative to the standard stochastic setting, we can still characterize reasonable convergence results of $\alpha_t$.
> Proving similar results for other notions of data dynamics is an interesting direction for future work.
>
> > discuss the pareto frontier of the trade-off between these two factors
>
> We appreciate your question on the tradeoffs between specificity and efficiency, particularly in the context of refresh prediction, and agree with the need to clarify this point in the revised manuscript.
> By construction, refresh prediction will have less specificity than the intersection-based method, as it only uses the prediction set at the final timestep; in contrast, the intersection-based method can only produce successively smaller prediction sets at each time step.
> However, because the score function of the intersection-based method is defined as the maximum non-conformity score across all timesteps, it is generally much more conservative relative to the prediction sets produced by refresh prediction.
>
> To further clarify this point, we plot the Pareto frontier of the intersection-based method and refresh prediction in the revised manuscript across various choices of $\alpha$.
> These results are available in Appendix C.
> In fact, across different levels of target coverage, refresh prediction tends to improve the Pareto frontier relative OA alone.

---

### Official Review · Reviewer_SzHF · 2024-11-02

**Soundness:** 3
**Presentation:** 3
**Contribution:** 2
**Rating:** 5
**Confidence:** 3

**Summary:**

This paper presents conformal reasoning, a method to reason about uncertainty in interactive environments.

Conformal reasoning builds upon two key techniques: (1) adaptive conformal inference that performs online uncertainty set prediction in the presence of distribution shifts, and (2) refresh prediction that design a new score function.

Experiments on medical information seeking and embodied question answering show the proposed method is better than existing baselines.

**Strengths:**

- Well motivated problem of reasoning about uncertainty in interactive environments
- Good empirical performance
- Paper and method are well presented and easy to understand

**Weaknesses:**

- Major weakness is limited novelty in the methods. The adaptive conformal inference is a straightforward application of the previous work, and the refresh prediction, as presented in the paper, is merely a new score function. So in this perspective the methodological contribution is quite limited.

- The ACI framework is a bit outdated now in the literature of online conformal prediction (with distribution shifts). In addition, several issues are known about ACI, including the requirement of parameter tuning in the algorithm. There are several recent algorithms that are theoretically more sound than ACI, such as:
  - Bastani, Osbert, Varun Gupta, Christopher Jung, Georgy Noarov, Ramya Ramalingam, and Aaron Roth. "Practical adversarial multivalid conformal prediction." Advances in Neural Information Processing Systems 35 (2022): 29362-29373.
  - Bhatnagar, Aadyot, Huan Wang, Caiming Xiong, and Yu Bai. "Improved online conformal prediction via strongly adaptive online learning." In International Conference on Machine Learning, pp. 2337-2363. PMLR, 2023.
  - Zhang, Zhiyu, David Bombara, and Heng Yang. "Discounted Adaptive Online Learning: Towards Better Regularization." In Forty-first International Conference on Machine Learning.
and the references therein.

- In the limitations section, the paper mentioned

> convergence and stability of Conformal Reasoning is dependent on the choice of learning rate in adapting $\alpha_t$, requiring some hyperparameter tuning for the best performance

However, there are no ablation studies on the sensitivity of hyperparameter tuning. If the performance of the algorithm is very sensitive to the hyperparameters, then it is unclear if the proposed method is actually better than baselines in a statistical sense.

- It should be noted that the coverage guarantee is "asymptotic", and we do not when the coverage guarantee starts to hold, when using conformal prediction in the presence of distribution shifts.

**Questions:**

Ablation studies are needed to study the sensitivity of hyperparameters on the performance of the algorithm.

---

> ### Author Response · Authors · 2024-11-24
>
> We appreciate your review and hope to clarify your concerns in the comments below and in our revised manuscript.
>
> > Major weakness is limited novelty in the methods
>
> While we agree that adaptive conformal inference is not new, to our knowledge, its application as a method for selecting $\hat q$ on data collected *offline* is novel, both in standard settings for conformal prediction as well as in interactive settings.
>
> - Indeed, we are not aware of other work that applies adaptive conformal inference on a calibration set to learn a reasonable $\alpha_t$ for the agent to use at test time; in turn, this helps eliminate some of the dependencies seen similar work in interactive settings that overly rely on human-in-the-loop feedback, such as [1].
> Additionally, the use of adaptive conformal inference to learn $\alpha_t$ enables our approach to overcome the presence of non-i.i.d. data in the calibration set in interactive reasoning tasks.
> We emphasize this point more clearly in the revised manuscript.
>
> - Second, while we agree that refresh prediction is not in itself novel --- indeed, it is one particular formulation for the score function --- its application is only enabled by using online conformal prediction techniques for offline calibration.
> Moreover, empirically, its application in interactive settings enables stronger performance in a number of metrics relative to the intersection-based method while maintaining the same coverage guarantees.
>
> - As mentioned in the overall response and reflected in the revised manuscript, we also introduce Guided Reasoning, which enables the use of the conformal prediction set within the agent's exploration process, as well as a third benchmark, Twenty Questions, to further demonstrate the benefits of our approach.
> In particular, we demonstrate that our approach enables the use of the conformal prediction set within the agent's policy, which to our knowledge has not been done in similar work.
> Moreover, the superior performance of Guided Reasoning relative to split conformal prediction emphasizes the utility and novelty of our approach.
>
> - Finally, we introduce several metrics beyond coverage and set size that help characterize the usefulness of our approach; in particular, these metrics (\% answered, efficiency) are unique to interactive settings instead of single-turn settings typical of conformal prediction.
> These metrics more broadly are useful for evaluating the success of different types of conformal algorithms in interactive settings.
>
> > ACI framework is a bit outdated now in the literature
>
> We agree that ACI is relatively outdated given recent work (especially those cited by the reviewer). However, in our work, we sought to use ACI as a simple, understandable, and easy-to-implement demonstration of our approach's benefits.
> Due to the nature in which our approach "consumes" the ACI method, other adaptive and online learning methods can easily be swapped in.
> Indeed, as a testament to this fact, we provide additional results using the adaptive frameworks from Zhang et al. (2024) and Bastani et al. (2022) in Appendix C of our revised manuscript on the EQA task.
> Despite the differences in these methods---for instance, minimizing different notions in regret, or striving for multicalibration---these methods are similarly effective in our problem setting.
> An interesting question for future work is characterizing the relationship between different types of online convex optimization approaches and their effectiveness in learning $\alpha$ from offline data; this question inherently is tied to online algorithms that can achieve "best of both worlds" guarantees on stochastic and adversarial data.
>
> > If the performance of the algorithm is very sensitive to the hyperparameters, then it is unclear if the proposed method is actually better than baselines in a statistical sense
>
> To further clarify the importance of the step size decay hyperparameter, we demonstrate the performance of our proposed approach against the step size decay hyperparameter in the revised manuscript.
> First, different choices of step sizes yielded similar terminal values of $\alpha_t$, indicating that our problem setting is robust given reasonably selected step sizes.
> Second, ablating the rate of decay for our learning rate similarly suggests flexibility in choosing the decay; nonetheless, selecting decay rates that are too small or too large suffer the equivalent variability of ACI or static nature of traditional offline conformal methods.
> Finally, the success of different types of online conformal prediction methods, which in general are more robust to the hyperparameters that induce variability in ACI, similarly accentuates the robustnesss of our approach.
>
> [1] Ren AZ, Dixit A, Bodrova A, Singh S, Tu S, Brown N, Xu P, Takayama L, Xia F, Varley J, Xu Z. Robots that ask for help: Uncertainty alignment for large language model planners. arXiv preprint arXiv:2307.01928. 2023 Jul 4.

---

> > ### Author Response · Authors · 2024-11-24
> >
> > > We do not know when coverage guarantee begins to hold
> >
> > While guarantees for convergence of $\alpha_t$ are asymptotic, we provide demonstrations of the empirical convergence of $\alpha_t$ over the course of the interaction in Appendix C of the revised manuscript.
> > While these results are not a formal proof of convergence and should not be interpreted as such, they demonstrate that the $\alpha_t$ selected by our approach is a reasonable heuristic for the agent to use at test time.

---

> > > ### Comment · Reviewer_SzHF · 2024-11-26
> > > **Thanks for the rebuttal**
> > >
> > > I appreciate the authors' response and new experiments.
> > >
> > > I have increased my rating to 5 (marginally below acceptance). My main justification is that the contribution of this paper, while good, is still a bit incremental. I am unclear if it meets the bar of ICLR, which appears to be very competitive.

---

### Official Review · Reviewer_1c1u · 2024-11-04

**Soundness:** 2
**Presentation:** 2
**Contribution:** 2
**Rating:** 3
**Confidence:** 4

**Summary:**

The paper introduces a method called conformal reasoning. It extends the algorithm of Ren et al. to an online, multi-turn setting, and makes the set construction more flexible (no more running intersection needed) by applying theory from adaptive conformal prediction. Experiments are shown on medical diagnosis and question answering datasets.

**Strengths:**

* The english prose is written relatively well, and it was easy to follow.
* I believe this problem setting is interesting and important to solve.
* Lifting the restriction of needing to take the running set intersection makes the algorithm of Ren et al. more practical.

**Weaknesses:**

(1)
There are fundamental and serious issues with the idea of terminating when |C| = 1, as proposed in Figure 1. To be fair, these problems appear not just in this paper, but in the prior work of Ren et al. as well.   The issue is this: marginal coverage does _not_ guarantee coverage when |C|=1. The latter is a form of conditional coverage. Thus, on the termination event in Figure 1, the coverage guarantee provided by conformal prediction does _not_hold. That is, precisely on the event that the reasoning is deemed reliable (|C|=1), we cannot provide a reliability guarantee. (Formally, $\mathbb{P}(y \in C(x) \mid \|C(x)\| = 1)$ has no guarantee.)

Conformal prediction methods have been developed for abstention, such as the conformal alignment algorithm of https://arxiv.org/abs/2405.10301, or the conformal selective classification subroutine of https://arxiv.org/abs/2110.01052. Such procedures _are_ able to guarantee $\mathbb{P}(y \in C(x) \mid \|C(x)\| = 1) \geq 1-\alpha$.

(2)
The mathematical details of the paper are unclear, largely due to many typos. It is also overly formal in some sections, while being under-developed in critical parts.

The biggest issue is on page 7, where the score function is defined. We see $s(\bar x, y) = s(x_i^{min(t_{ans},T)},y)$. But $t_{ans}$ is unknown at $t=1$; so how can we expect to construct the prediction set at time $t=1$? That is: we can obviously define any score function we would like, as a function of the whole trajectory. The harder part, at least in this multi-turn setup, is inverting the test to get a sensible prediction set. Therefore, the form of the set should be written out, as it is for the baseline at the bottom of page 5.

There are also a number of typos in the math throughout. My general suggestion would be to cut down on the amount of notation and formalism as much as possible, to reduce to the minimal amount needed to convey the ideas.

Math typos:
* Bottom of page 5, definition of sets: $C^{\tau}$ does not seem to depend on $\tau$ except through $x^{\tau}$, which calls into question why it should have the second superscript.
* in the definition of SCP on page 5, the quantile function has been defined over measures, but it takes an empirical PDF as an argument earlier (these objects are of a different type)
* the conformal quantile isn’t the 1-alpha empirical quantile, it is the$ (1-\alpha)(1+1/n)$ empirical quantile due to the delta mass placed at infinity
* The correct notation for a set cardinality is not $\| \mathcal{C} \|$, but $|\mathcal{C}|$ (the former is normally reserved for a norm)

(3)
The paper could also use a more comprehensive literature review around methods for conformal prediction under feedback loops, for agents, and for decisions. There are many such papers.

_______

Other typos, errors, and minor suggestions:

$\|y - f(x)\|$ is not bounded. (Conformal scores do not generally need to be bounded or non-negative.)

“originally designed to handle external distribution shifts in dynamic environments” not really. It was designed to handle arbitrary distribution shifts of any kind.

Figure 1: right-side of the right-most panel, I think that there is an error. The algorithm shouldn’t yet terminate because both A and B cross the red line, so the prediction set should be {A, B}. Maybe the bar for A should be lower.

“minimal prediction sets” -> “minimal size prediction sets”

“We emphasize that Theorem 1 holds for arbitrary score function” typo

“The main challenge of the task is for the Expert system to ask
information-seeking questions to expand Kt until the the Expert system is sufficiently” typo in Appendix A.

On pg. 7, why does $x_i$ appear in the definition of $s(\bar x, y)$? This must be a typo, given the way $\bar x$ is defined on pg 5 in terms of $x^i$.

Not sure why this is called conformal reasoning. Name does not reflect the nature of the method, which can be applied to any trajectories, and does not really include reasoning as a subroutine.

**Questions:**

Main question: how is the set constructed when inverting the score on pg. 7?

---

> ### Author Response · Authors · 2024-11-24
>
> Thank you for your thoughtful and detailed review. We greatly appreciate your recognition of the clarity and importance of our problem setting and your constructive comments on the theoretical and methodological aspects of the paper. Below, we address each of your points carefully and outline the changes made to the manuscript in response to your feedback. We hope that these revisions address your concerns and encourage you to raise your score for our work.
>
> ### **Theoretical Limitations of Termination at $\left| C\right| = 1$**
>
> We fully agree with your observation that the marginal coverage guarantee does not apply when the prediction set size is exactly 1. As you noted, this is indeed a limitation of our approach as well as of related prior works (e.g., Ren et al.). In the revised manuscript, we explicitly clarify that our approach does not seek to achieve conditional coverage for such cases. Instead, we emphasize that our goal is to guarantee *marginal coverage* over the prediction set at termination.
>
> **Key points addressed in the revision:**
>
> - We provide additional discussion in Section 5 to highlight that our empirical results demonstrate strong marginal coverage even when $\left|C\right|$ = 1.
>
> - We contrast our approach with the hypothesis testing-based methods you suggested (e.g., Conformal Alignment), which have distinct objectives and naively do not produce actionable prediction sets at termination time. By allowing predictions to terminate at $|C| = 1$, we offer a more practical approach that enables higher efficiency (albeit, with no guarantees on the "correctness" of the prediction) than continuing to the maximum sequence length as well as retaining the utility of outputting a prediction set.
>
> > Conformal prediction methods have been developed for abstention
>
> The two hypothesis testing-style approaches mentioned in the review are an interesting alternative to our approach, and we agree that this is an interesting direction for future work.
> However, a notable limitation of these hypothesis testing approaches is that naively controlling the selective classification error does not yield a meaningful prediction set.
> In particular, in both Learn-then-Test and Conformal Alignment, the authors compute conformal $p$-values for each test point and use these $p$-values to determine whether the model's prediction should be used and elicited or abstained from.
> Accordingly, a hypothesis testing approach to our set of interactive tasks would not lead to prediction sets, but it is interesting to make the connection in future work.
> In contrast, a prediction set-based method like ours offers a terminal prediction set that can be used to aid a human practitioner that may be more useful in practice.
>
> Our approach, to a limited extent, *interpolates* between these two approaches.
> In particular, for Ren et al. (2024) to achieve a correct $1-\alpha$ coverage guarantee, their construction requires them to terminate at the maximum allowed timestep for their specific scenario; while not achieving conditional coverage, this algorithmic approach would provide a marginal guarantee, at the expense of the efficiency at test time.
> In contrast, the aformentioned risk control approaches ensure that terminating at a prediction set of size one would provide a valid conditional coverage guarantee, but at the cost (in a naive implementation) of not offering any terminal prediction set that might aid a human practitioner.
> Here, we empirically demonstrate that we *do* in fact retain marginal coverage for the final prediction set at termination time, whether that be because of the prediction set size or the maximum sequence length.
> Moreover, our approach offers a more efficient alternative to the approach of Ren et al. that goes to the maximum sequence length, as we enable the agent to terminate at a prediction set of size one when appropriate.
> Finally, should the agent not be able to terminate at a prediction set of size one, our approach still offers a terminal prediction set that can be used to aid a human practitioner.
>
> We appreciate the feedback and have added the above discussion to the end of Section 2 of the revised paper.
>
> > The mathematical details of the paper are unclear, largely due to many typos. It is also overly formal in some sections, while being under-developed in critical parts.
>
> We appreciate your detailed feedback on mathematical clarity and the noted typographical errors. In the revised manuscript, we have simplified and clarified the mathematical descriptions, particularly in Section 4, to reduce unnecessary formality while maintaining rigor, and we have corrected all identified typos.

---

> ### Author Response · Authors · 2024-11-24
>
> ### **Literature Review on Feedback Loops and Decision-Making**
>
> We expanded our literature review on methods for conformal prediction under feedback loops and decision-making. In particular:
>
> - We now incorporate a broader review of conformal prediction techniques applied to interactive tasks, including recent works on feedback shifts and trajectory-specific methods.
> - We additionally characterize the broader literature of online conformal approaches and our relationship to their empirical and theoretical characterizations.
> - We emphasize how our approach builds upon these foundational methods to address unique challenges in dynamic, multi-turn environments.
>
> These additions, included in **Section 2**, highlight the broader significance of our work and its relationship to the state of the art.
>
>
> ### **Other Minor Suggestions and Typographical Errors**
>
> We appreciate your detailed feedback on smaller issues and have incorporated your suggestions:
>
> - Updated “minimal prediction sets” to “minimal size prediction sets.”
>
> - In **Figure 1**, the termination criteria demonstrated is the traditional criteria without refresh prediction -- this means that if an option has ever been eliminated in a previous turn (e.g. A was below the red line in turn 1), then it cannot be "recovered" even if its probability exceeds the threshold in a future turn. This is also one of our motivations to propose the Refresh Prediction technique to allow for more fault-tolerant set prediction. We have updated the caption to clarify this point.
>
> - Addressed the inconsistency in how $\bar{x_i}$ appears in the score function definition, as it was indeed a typo.
>
> - Provided a brief justification for calling our method “Conformal Reasoning,” emphasizing its novel combination of prediction set construction and interactive decision-making.
>
>
> We have significantly revised the manuscript to address the issues you raised, including clarifying theoretical limitations, improving mathematical formalism, expanding the literature review, and correcting typographical errors. Furthermore, we have clarified the distinctions between our work and alternative approaches (e.g., hypothesis testing methods), highlighting the practical benefits of our approach in real-world scenarios.
>
> We kindly request you to consider these improvements when reassessing your evaluation of the manuscript. Your feedback is extremely invaluable, and we thank you again for helping us refine and improve this work.

---

> > ### Comment · Reviewer_1c1u · 2024-11-25
> >
> > I have a quick question: on the right hand side of Figure 1, it says “termination criterion: $\|C\|=1$.”
> >
> > Firstly, the typo in the norm vs set cardinality remains. Secondly, I am confused as to what the role of the termination criterion is. My main worry is that, if your goal is to take some action (with guarantees) on the event that the cardinality is 1, those guarantees would be invalidated by the loss of conditional coverage, as we discussed.

---

> ### Author Response · Authors · 2024-11-27
>
> > Firstly, the typo in the norm vs set cardinality remains.
>
> So sorry -- we were initially confused when you mentioned the issue with using the norm vs. set cardinality and looked for it in the text; it didn't occur to us that the issue was in the figure. We have fixed that in the current revision.
>
> > Secondly, I am confused as to what the role of the termination criterion is. My main worry is that, if your goal is to take some action (with guarantees) on the event that the cardinality is 1, those guarantees would be invalidated by the loss of conditional coverage, as we discussed.
>
> Regarding conditional guarantees and the termination criteria, we hope to clarify that our goal is to **not** to ensure conditional coverage if $|C|=1$; rather, we aim to provide **marginal coverage** across our entire algorithmic process.
> Our termination criteria is empirically motivated: we would ideally like to end exploration as long as the model is reasonably confident in the correct answer.
> In particular, for a given scenario and trajectory $(x_\xi^0,\dots, x_\xi^{t})$, where $t=\min(T, t_{ans})$, we hope to provide achieve coverage
> $$
> P(y_\xi \in C(x_\xi^t))\geq 1-\alpha
> $$
>
> Accordingly, our guarantee is marginal over our entire collection of trajectory termination times $t_\xi$.
> Additionally, we emphasize that the algorithmic approach used for learn $\alpha_t$ is exactly that used at test time; that is, they have the **same** termination criterion.
>
> Moreover, consider the coverage guarantee from Ren et al. if done correctly, that is, if they let each test sequence run to maximum scenario length.
> Their failure to achieve a coverage guarantee comes from the fact that if they continued beyond the timestep at which they get |C|=1, their conformal set could be entirely empty, which would not cover the true labely $y$.
> As a result, the SCP approach from Ren et al. overcovers relative to the actual target coverage level.
> Alternatively, in our approach, by explicitly adapting $\alpha_t$ to whether coverage was attained -- that is, if we overcover at that time step, we will make $\alpha_{t+1}$ **larger** and the quantile "less restrictive" -- we are able to better mitigate the overcoverage phenomenon present in SCP.

---

### Author Response · Authors · 2024-11-24
**Summary Rebuttal**

We appreciate the reviewers' thoughtful questions regarding our submission.
We're grateful that many of the reviewers agreed with the significance and motivation of our problem setting, our strong empirical performance, and the overall practicality of our method.
Here, we highlight important changes in our revision, as well as address common questions that arose in the review process. Reviewer-specific questions will be addressed individually.

**Important Updates (highlighted in blue text in the revised paper)**

1. **Naming**: We realized that the confusion brought up by some of the reviewers regarding novelty is due to the overloading of the term "adaptive conformal inference (ACI)." Although our method builds upon techniques from online conformal prediction like ACI, it is fundamentally different as it *learns* the conformal threshold *offline*. This differs from ACI by eliminating the need for sequential feedback, i.e. receiving $Y_{t-1}$ before $X_t$, and making it possible for real-world deployment in settings where a ground-truth label is not readily available (e.g. medical diagnosis).  Therefore, we rename the component of our proposed method **"offline adaptation"**, which adapts high-level intuition from ACI to account for distribution shift. Moreover, our approach is compatible with more general types of online conformal methods; however, we largely appeal to ACI for its theoretical characterization in both adversarial and stochastic settings.

    - Additionally,we agree with some reviewers that Refresh Prediction can be viewed as an alternative score function, but it is only enabled through the use of offline adaptation; in particular, it accentuates that the proposed adaptive offline method is robust to flexible score functions and set constructions in contrast to standard split conformal methods.

2. **Guided Reasoning**:
To demonstrate the theoretical flexibility of our offline adaptation approach, we show that it is robust to both 1) novel score functions and 2) temporally dependent reasoning paths by proposing Refresh Prediction and Guided Reasoning, respectively.
We add Guided Reasoning to our revision to show that the robustness of offline adaptation is not limited to switching score functions but rather a more fundamentally wide set of modifications.
In contrast, SCP does not enable either Refresh Prediction because of the restrictiveness in its construction, nor can it leverage Guided Reasoning because using the prediction set in the subsequent reasoning path would break the independent assumption.

3. **20 Questions**: We have strong empirical evidence that the proposed method improves model performance on critical, high-stakes tasks such as clinical reasoning and embodied question answering. In order to strengthen our results in response to some of the reviewers' concerns regarding performance and trade-offs between specificity and efficiency, we have added the 20 Questions task to demonstrate the effectiveness of Conformal Reasoning (and in particular, Guided Reasoning), especially in settings where the size of the label set is large.
Moreover, our results indicate that the tradeoff between efficiency and specificity is task-dependent.
We explore these results further in Section 7 of the revision.

Below are our results on the Twenty Questions task, and we also show that GR significantly improves accuracy in Figure 3 of the revised paper.

| Target Coverage | Refresh Prediction | Guided Reasoning | Empirical Coverage | \% Answered ↑ | Specificity (\%) ↑ | Efficiency (\%) ↑ |
|--|--|--|--|--|--|--|
| 0.9 | ✗ | ✗ | 90.5 ± 1.1 | 0.1 ± 0.2 | 52.6 ± 1.4 | 4.0 ± 0.4 |
|  | ✓ | ✗ | 88.4 ± 2.0 | 5.7 ± 1.4 | **67.6** ± 1.4 | 7.9 ± 0.8 |
|  | ✗ | ✓ | 92.3 ± 1.9 | 2.5 ± 1.6 | 59.3 ± 2.9 | 6.5 ± 0.8 |
|  | ✓ | ✓ | 92.0 ± 2.0 | **8.1** ± 1.6 | 63.4 ± 4.5 | **10.2** ± 1.5 |
| 0.8 | ✗ | ✗ | 83.2 ± 3.1 | 0.5 ± 0.4 | 62.4 ± 2.1 | 4.6 ± 0.5 |
|  | ✓ | ✗ | 82.7 ± 4.6 | 10.8 ± 2.1 | 77.5 ± 2.5 | 8.9 ± 1.1 |
|  | ✗ | ✓ | 82.9 ± 2.0 | 3.8 ± 1.0 | 68.7 ± 1.0 | 7.3 ± 0.6 |
|  | ✓ | ✓ | 79.2 ± 3.9 | **16.0** ± 3.3 | **79.5** ± 3.5 | **12.4** ± 0.5 |
| 0.5 | ✗ | ✗ | 53.4 ± 3.8 | 2.5 ± 0.5 | 79.5 ± 0.4 | 4.8 ± 0.5 |
|  | ✓ | ✗ | 52.5 ± 3.6 | 38.9 ± 4.5 | 93.7 ± 2.0 | **14.3** ± 1.2 |
|  | ✗ | ✓ | 54.5 ± 3.4 | 15.2 ± 3.0 | 86.0 ± 0.4 | 7.6 ± 1.0 |
|  | ✓ | ✓ | 50.3 ± 5.0 | **43.0** ± 2.8 | **94.0** ± 0.7 | 13.2 ± 1.0 |

---

### Meta-Review · Area_Chair_UB9z · 2024-12-19

**Metareview:**

The paper 'Conformal Reasoning: Uncertainty Estimation in Interactive Environments' was reviewed by 4 reviewers who gave it an average score of 4.75 (final scores: 3+5+5+6). The reviewers found the ideas rather well presented in the paper, the analysis insightful, and the experiments interesting. On the other hand, the novelty was seen as limited, the experiments should have covered more setups and ablations, and some parts of the paper were confusing. Overall, the reviewers see that the paper does not quite meet the bar for acceptance in its current form.

**Additional Comments On Reviewer Discussion:**

The authors provided a rebuttal that addressed some of the concerns by the reviewers. The discussion was not active for this paper and only two of the four reviewers participated in the author-reviewer discussion. The average score increased from 4.25 -> 4.75 during the discussion.

---

### Decision · Program_Chairs · 2025-01-22

Reject